# The Invisible Lottery: How Subtle Cues Steer Algorithm Choice in LLM Code Generation

**Akanksha Narula** [1]  **Mofasshara Binte Rafique** [2]  **Laurent Bindschaedler** [1]

## Abstract

Large language models (LLMs) now generate substantial production code, often for tasks with multiple valid algorithmic solutions. Incidental prompt cues, meaning contextual words or metadata outside the task specification, can steer *which* algorithm the model selects, even when all outputs pass the same tests. Prompt sensitivity is well studied as a tool to improve output quality. Here, output policy means algorithm choice under fixed correctness. We define algorithm steering as cue-induced shifts in algorithm-family distributions and run 46,535 controlled experiments across 11 tasks, 19 cue types (18 channels plus a memoization semantic-vs-surface ablation that preserves meaning while changing typography and punctuation), and 15 model configurations. We find large, systematic shifts in algorithm-family distributions (up to 100 pp), largely consistent with cue semantics, including in applied tasks such as rate limiting. Direct algorithm naming is the most reliable mitigation we tested. Accidental context therefore creates an "invisible lottery" over performance, security, and maintainability.

## 1. Introduction

Modern artificial intelligence (AI)-assisted development has a simple workflow in which developers write a specification, receive an implementation, run the tests, and ship if they pass (Barke et al., 2023; Mozannar et al., 2024). Passing tests only confirms functional correctness. When multiple algorithms solve the same problem, the model must choose among them, and this choice is invisible to the developer. A Fibonacci function might use naive recursion ($O(2^n)$) or matrix exponentiation ($O(\log n)$). Both pass small-input

tests, such as inputs where $n$ is no larger than 20, and diverge at scale. The difference determines whether the code scales to production or fails on large inputs, yet the developer may never know which algorithm they shipped.

Context drives the model's choice. Modern AI coding assistants have access to system prompts, surrounding code, and project metadata. We call these contextual elements *cues*. These cues can tip the model toward one algorithm or another, while developers have little visibility into this algorithm-selection process and lack obvious audit tools. Generated code therefore embeds an implicit policy of performance characteristics, memory footprints, and maintenance burdens that correctness-based evaluation misses.

Prompt sensitivity in LLMs is well studied as a *tool*. Researchers craft prompts to elicit desired behaviors, anchor models to user intent, or improve security (Lam et al., 2025; Tian & Zhang, 2025; Tony et al., 2025). These lines of work treat prompts as a control surface for output *quality*—whether the model succeeds or fails. We study output *policy*: which valid solution the model selects when multiple exist. Correctness-based benchmarks such as HumanEval and MBPP (Mostly Basic Python Problems) miss this variation by design, since pass@$k$ collapses every passing implementation into a single bit; the consequences for performance, security, and maintainability can be substantial.

Prompt context systematically steers algorithm selection, even when cues lack explicit algorithmic meaning. We use *algorithm steering* for shifts in algorithm-family distributions, the classes of implementation strategies among functionally correct outputs. In 46,535 controlled experiments on 8 classical and 3 applied tasks, we find steering deltas of up to 100 pp, far larger than typical prompt sensitivity effects (Sclar et al., 2024). Cues we designed as neutral placebos (team names, project codes, color themes) still produced a mean 26 pp shift; across all conditions, individual cells reach 100 pp, even the *presence* of random identifiers constrains algorithm choice toward conventional solutions relative to a no-cue baseline.

The largest swings match cue semantics in several tasks. For tree traversal, recursion ranges from 0% under a space cue to 89% under readability (89 pp swing), without affecting cor-

[1]Max Planck Institute for Software Systems (MPI-SWS), Saarbrücken, Germany [2]Independent Researcher, Switzerland. Correspondence to: Akanksha Narula <anarula@mpi-sws.org>.

*Proceedings of the 43rd International Conference on Machine Learning*, Seoul, South Korea. PMLR 306, 2026. Copyright 2026 by the author(s).

rectness. A `prototype` context activates `eval`[1] shortcuts in 70% of expression-parsing outputs vs. 6% under `interview` (64 pp). A `junior` persona yields space-optimized memoization (100%) vs. 14% under `academic`. For moving sum, sliding-window implementations rise from 47% under correctness to 97% under time. Figure 1 summarizes these representative steering effects.

The effects are also model-dependent. The same `academic` persona causes Claude Sonnet 4 to select matrix exponentiation and pass all tests, while GPT-5 attempts matrix exponentiation in 4 of 5 runs and fails them all (0% pass rate). Gemini 3 Flash avoids the failure mode by defaulting to naive recursion. Switching models changes which algorithms a cue activates and whether the code works.

These results frame LLM code generation as an *invisible lottery*. Real prompts carry context (system prompts, project metadata, import statements), and this context biases algorithm selection in ways developers neither intend nor detect. The lottery has two stages: the cue determines which algorithm the model attempts, and the model determines whether that attempt succeeds. Tests confirm correctness but rarely reveal whether memory grows as $O(n^2)$ when $O(1)$ was possible, or whether call-stack depth grows with input size. Algorithm choice must therefore be evaluated alongside correctness: performance, security, and maintainability can shift without failing tests. We study this through four research questions on cue values, channel strength, model/temperature variation, and sophistication tradeoffs.

**Contributions:**

- **Methodology**. We build an abstract syntax tree (AST)-based detection framework and a benchmark suite of 11 multi-algorithm tasks (8 classical + 3 applied) (Section 3).
- **Phenomenon**. We identify and quantify algorithm steering, with up to 100 pp effects across 19 cue types, 11 tasks, and 15 model configurations, and provide a taxonomy of steering channels (Section 4).
- **Actionable insight**. "Expert" personas reduce reliability where sophisticated algorithms introduce failure modes (e.g., matrix exponentiation on memoization); direct algorithm specification restores it (Section 5).

## 2. Background and Problem Formulation

### 2.1. Problem Setting

Let $\mathcal{T}$ denote an evaluation task (with tests) and $\mathcal{A}_\mathcal{T} = \{a_1, \ldots, a_k\}$ the set of algorithm families satisfying it. A model $M$ with prompt $p$ under fixed decoding $\theta$ induces $P_{M,\theta}(a \mid \mathcal{T}, p)$ over $\mathcal{A}_\mathcal{T}$; we estimate this empirically

---

[1] `eval` is a built-in function that executes a string as code; it is brittle and can be unsafe if inputs are adversarial.

from repeated API calls for fixed $(M, \mathcal{T}, p, \theta)$, matching the distribution an API user observes. We decompose $p = (c, s)$ into cue $c$ and specification $s$. The *steering effect* of $c$ relative to baseline $c_0$ is $\Delta(c, c_0, a) = P_{M,\theta}(a \mid \mathcal{T}, c, s) - P_{M,\theta}(a \mid \mathcal{T}, c_0, s)$, measured in percentage points (pp). A cue steers when the absolute value of $\Delta$ exceeds $\epsilon$; we use $\epsilon = 15$ pp (Appendix A.7). Channel-level *steering capacity* is $\max_{a,c,c'} |\Delta(c, c', a)|$. We use $T=0$ to remove sampling variability; many users expect $T=0$ to be deterministic, but across 5,674 API cells at $T=0$ (typically 5 reps each), 1,197 (21.1%) produce multiple distinct labels (`unknown` included) and 936 (16.5%) produce multiple distinct classified algorithm families across replications. The cue effect persists across temperatures 0, 0.3, 0.7, and 1.0 at high pass rates (Appendix A.8).

### 2.2. The Algorithm Selection Problem

The algorithm selection problem (Rice, 1976) asks which algorithm from a portfolio is best suited for a given problem instance. Many programming tasks admit multiple valid solutions. A correct Fibonacci implementation may use naive recursion ($O(2^n)$ time, $O(n)$ space; subsequent pairs use this time/space notation), memoized recursion ($O(n)/O(n)$), bottom-up dynamic programming (DP, $O(n)/O(n)$), space-optimized DP ($O(n)/O(1)$), or matrix exponentiation ($O(\log n)/O(1)$).

All produce identical outputs for valid inputs, yet differ in computational properties and suitability for different contexts. The choice is a *policy decision* reflecting implicit assumptions about problem context. Human programmers consider input scale, memory constraints, performance requirements, and maintainability, but specifications rarely state these factors explicitly.

The same ambiguity appears in other domains:

- Substring search can use naive scanning, Knuth–Morris–Pratt (KMP), or Boyer–Moore.
- Topological sort can use depth-first search (DFS) or Kahn's algorithm.
- Shortest paths can use breadth-first search (BFS, unweighted/unit weights) or Dijkstra.
- Tree traversal can be recursive, iterative, or Morris.

All are correct, but each encodes different tradeoffs.

### 2.3. Steering Channels

We define a *steering channel* as any prompt feature that influences algorithm selection without explicitly specifying an algorithmic approach. Table 1 (left) lists 8 channel categories and example cue values; we evaluate 18 channel types, each with multiple cue values, and Figure 2 reports within-channel steering. The full mapping from channel types to cue values appears in Appendix A.1 (Table 3).

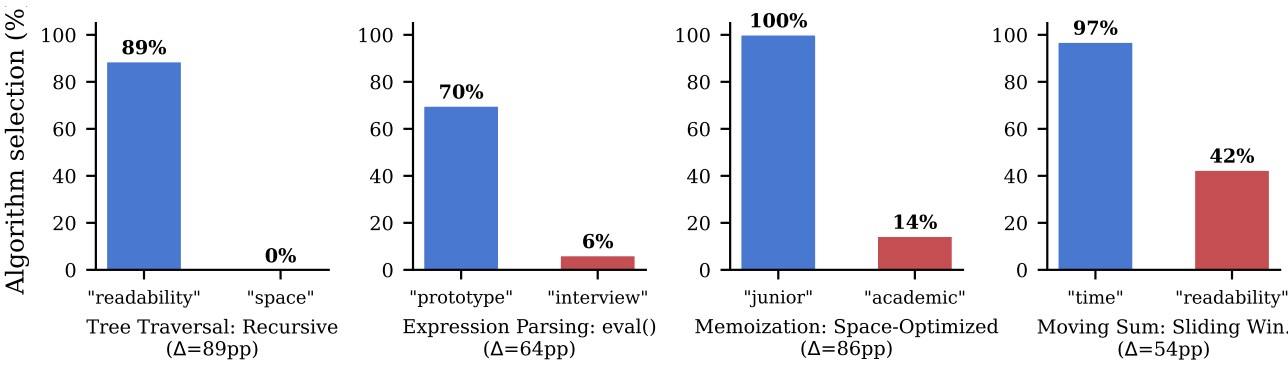

*Figure 1.* Four representative steering cases. Each group holds the task specification fixed and compares the max–min cue pair within that task–channel; bars report the selected algorithm-family rate.

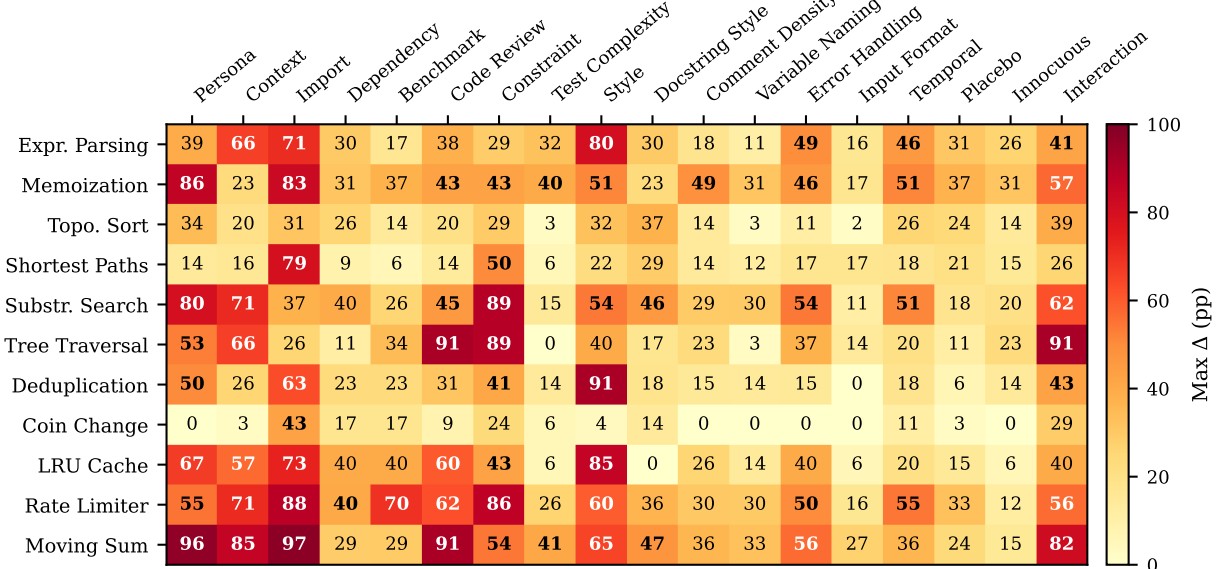

*Figure 2.* Maximum steering delta (pp) per task–channel pair. Each cell is the largest algorithm-family swing observed when varying cue values within that channel. Appendix Figure 5 reports the baseline-referenced version.

Each channel corresponds to a minimal cue injection with a fixed template. Some channels encode optimization priorities that are semantically meaningful without specifying a concrete algorithm. Steering deltas define the taxonomy.

**Scope and Non-Goals**   We study algorithm selection in classical tasks and add three applied tasks (least-recently-used (LRU) caching, rate limiting, moving-sum aggregation) to probe transfer. API/version selection, formatting, architecture, and multi-file generation lie outside scope.

## 3. Methodology

### 3.1. Task Suite

We use 11 algorithmic tasks spanning classical algorithms and three applied tasks (Table 1, right). Each task includes a signature and tests without algorithmic guidance; the constraints make all listed families valid, and the classifier excludes nonconforming algorithms. Full task signatures and family definitions are in Appendix A.2.

**Applied Task Semantics**   The rate limiter implements a sliding-window log: a request at time $t$ is allowed iff the number of *previous* requests in $(t - \texttt{window\_s}, t)$ is below `max_requests`. The moving-sum task returns sums of all length-$k$ windows.

**Performance Guardrails**   We add two stress tests outside functional correctness: Fibonacci with $n=36$ (1.0s timeout) and a skewed tree of depth 2,000. Across tasks, 735 of 2,775 naive recursive implementations (26.5%) hit guardrails, while iterative approaches pass at much higher rates. Detailed guardrail breakdowns are in Appendix A.10.

| Steering channels | | Task suite | |
|---|---|---|---|
| **Category** | **Example values** | **Task** | **Families** |
| Persona | academic, competitive, junior | Expr. Parsing | rec. descent, shunting, precedence, `eval` |
| Context | production, interview, prototype | Memoization | naive rec, memo, DP, space-opt, matrix exp |
| Import | `functools`, `typing`, `collections` | Topo. Sort | DFS, Kahn |
| Constraint | time, space, readability | Shortest Paths | Dijkstra, Bellman–Ford, BFS (unit) |
| Style | docstring, naming, error handling | Substring | naive, KMP, Boyer–Moore, Rabin–Karp |
| Format | json, doctest, table | Tree Traversal | recursion, stack, Morris |
| Temporal | 2008, 2024, legacy | Deduplication | `set`, `dict.fromkeys` |
| Control | placebo, interaction, innocuous | Coin Change | DP (top/bot), BFS |
| | | LRU Cache | dict+DLL, `OrderedDict` |
| | | Rate Limiter | scan, window, bucket |
| | | Moving Sum | naive, prefix, window |

*Table 1.* Steering channel taxonomy (left) and task suite with valid algorithm families (right). Example values are non-exhaustive.

## 3.2. Experimental Design

For each (task, cue, model) combination, we construct a prompt, generate code, run tests, and classify the algorithm family via AST analysis.

**Models and replication**   We evaluate 15 configurations. *API*: GPT-5, GPT-4o, Claude Sonnet 4, Gemini 3 Flash, GLM-4.7. *Local*: DeepSeek-Coder v2 16B, Llama 3.3 70B, Qwen2.5-Coder 32B/7B, DeepSeek-R1 70B, Devstral. *Quantized*: DeepSeek-R1 70B Llama-distill (FP16, Q8_0), Qwen2.5-Coder 32B Instruct (FP16, Q8_0). We run temperature sweeps for Qwen-32B and DeepSeek-R1 at 0.3, 0.7, and 1.0; all others use $T=0$. We replicate API models $5\times$ per (task, cue) cell and run local/quantized models one-shot; applied tasks (LRU, rate limiter, moving sum) appear in both the API and local one-shot grids. Inferential claims rest on the five replicated API models. The full experiment set comprises 47,075 runs. The main grid covers 46,535 runs over 1,135 conditions: $5 \times 1{,}135 \times 5 = 28{,}375$ API $+ 8 \times 1{,}135 = 9{,}080$ local $+ 2 \times 1{,}135 \times 4 = 9{,}080$ temperature-sweep, at $\geq 99.9\%$ completion (29 API runs short of the design). Three targeted studies extend the grid: 120 explicit-algorithm-specification runs (Section 5.6), 180 HumanEval steering runs (Section 4.8), and 240 high-replication robustness runs (Appendix A.9).

**Metrics**   We report algorithm-family distributions, steering deltas, and pass rates. We also track shortcut usage (e.g., `eval`) and distribution entropy as diagnostics.

**Statistical Analysis**   We report point estimates and effect sizes (max–min, mean, median, interquartile range (IQR)), grounded by a high-replication robustness check (240 runs, Appendix A.9), classifier-noise calibration ($\pm\sim 5$ pp, Ap-

pendix A.6), and a per-cell power analysis at $\epsilon=15$ pp (Appendix A.7). Appendix A gives the full measurement setup; Appendix B provides extended result tables. Algorithm-family percentages use the *classified denominator* (excluding `unknown` outputs); pass and unknown rates use the raw per-cell run count $N$ (typically 35 in the main grid).

## 3.3. Algorithm Detection

We use an AST-based classifier inspired by structural code representations (Alon et al., 2019) to map each generation to an algorithm family; we treat `eval/literal_eval` as shortcuts and additionally track `functools.lru_cache` and `OrderedDict` markers. We execute outputs against tests, so detected shortcut rates reflect runtime behavior, not keyword artifacts. Appendix A.4 summarizes the task-specific detection patterns. We label generations `unknown` when every family remains below 0.3 confidence; Appendix A.5 reports threshold sensitivity.

Classifier validation on 100 samples shows 87% overall agreement, with per-task accuracy ranging from ∼95% on tree traversal to ∼82% on expression parsing (topological sort, shortest paths, and memoization ∼90%). Disagreements concentrate where algorithm families share structural features (e.g., top-down vs. bottom-up DP) and so trade probability mass between structurally adjacent families rather than spreading uniformly. We additionally run an LLM-as-judge calibration on 110 stratified samples (GPT-4o judge): 80 exact matches, 17 label-granularity differences (both correct at different specificity), 8 ambiguous edge cases, and 5 genuine disagreements (∼5%). After normalizing for label granularity, agreement is 88%. A 13% random misclassification rate gives a standard error of $\approx \pm 5$ pp at $N=40$ (high-replication) and $\approx \pm 5.7$ pp at the main-grid

per-cell $N{=}35$, well below our 40 pp median effect. Full classifier-calibration details are in Appendix A.6.

### 3.4. Prompt Construction

Prompts combine a base task specification (constant within task), cue injection (varies by condition), and output format instructions; the bold line is the injected cue, and removing it yields the baseline. For example:

```
Performance Critical: Execution
speed is the top priority.
Implement inorder traversal of a
binary tree.
Signature: inorder(root: TreeNode)
-> list
Example: inorder(root) returns [1,
2, 3] for a simple tree.
```

Appendix A.3 lists the shared prompt structure and representative cue injections.

### 3.5. Evaluation Strategy

We separate channels by whether they carry algorithmic semantics. *Semantic* channels (constraint, persona, context, import) encode optimization priorities, role, project type, or library affordances; *innocuous* channels (team names, project codenames, color themes, random identifiers) are algorithmically irrelevant. Across 5 API models, semantic channels show more than 15 pp steering in 77.7% of cells (mean 67.2 pp, median 100 pp; $n{=}220$); innocuous channels do so in 48.2% of cells (mean 26.1 pp, median 0 pp; $n{=}110$). Semantic cues steer far more strongly, but nearly half of innocuous-channel cells still cross the 15 pp threshold. We run a focused persona×constraint interaction subset (memoization, tree traversal, rate limiter), report distributions both unconditional and conditioned on passing, and observe that the `none` baseline often yields *more* diversity than any identifier (memoization: 35% space-opt vs. 68–84% under placebos), so subtle context narrows choice.

## 4. Experimental Results

We ran 46,535 experiments across 11 tasks (8 classical, 3 applied), 19 cue types (18 channels plus a semantic-vs-surface ablation), 15 model configurations, and four temperatures. Overall pass rate is 91.7%, with 5.2% `unknown`. The results answer four research questions:

**RQ1** Do within-channel cue changes shift algorithm-family distributions under fixed specifications?

**RQ2** Which channels steer most strongly, and how consistently across tasks?

**RQ3** How do effects vary across models and temperatures?

**RQ4** How does correctness change when steering activates

| Model | Cells | Mean (pp) | Median (pp) | >15 pp | >30 pp |
|---|---|---|---|---|---|
| GPT-5 | 187 | 52.3 | 40.0 | 77% | 63% |
| GPT-4o | 187 | 38.1 | 0.0 | 49% | 42% |
| Gemini 3 Flash | 187 | 51.3 | 100.0 | 51% | 51% |
| GLM-4.7 | 187 | 51.7 | 50.0 | 80% | 64% |
| Claude Sonnet 4 | 182 | 56.5 | 80.0 | 66% | 60% |
| **All 5 API** | **930** | **50.0** | **40.0** | **65%** | **56%** |

*Table 2.* Per-model channel-level steering capacity across (task, channel) cells at $T{=}0$, five replications per cell, classified denominator. A cell is counted if at least two of its cue values each produced at least one classified output (otherwise the max–min is undefined). The last two columns report fractions of cells exceeding 15 pp and 30 pp. Cell counts are 187 for four of the five models; Claude Sonnet 4 has 182 because six cells (one on `lru_cache/docstring_style`, four on `rate_limiter` channels, and one on `shortest_paths/style_exemplar`) returned only `unknown` for at least one cue value.

more sophisticated algorithms?

### 4.1. Summary Findings

We test whether within-channel cue changes shift algorithm distributions (RQ1) and which channels steer most strongly (RQ2). For each (model, task, channel) cell we compute the steering capacity $\max_a \max_{c,c'} |P_{M,\theta}(a \mid \mathcal{T}, c) - P_{M,\theta}(a \mid \mathcal{T}, c')|$ in pp and summarize across cells. Figure 2 gives the task–channel matrix (Appendix Figure 5: baseline-referenced); Figure 3 gives representative before/after distributions; Appendix B contains full per-cue tables.

**Per-model effect sizes (API)** Across 930 qualifying API cells at $T{=}0$ (5 models × up to 188 cells, where $188{=}11$ tasks × 17 baseline-bearing channels + 1 memoization-only ablation; `input_format` has no baseline), the per-cell capacity has median 40 pp, mean 50.0 pp; 65% of cells exceed 15 pp and 56% exceed 30 pp. Four of five API models exceed 50 pp mean steering; GPT-4o is the outlier at 38.1 pp mean and 0 pp median, a bimodal pattern in which its algorithm choice is invariant on roughly half of cells (median 0) but shifts substantially on the remainder (mean 38.1; Table 2). We adopt mean/median/IQR as primary metrics; max–min is an auxiliary worst-case statistic.

**Results** Per-cell steering deltas reach up to 100 pp; pooling across the 15 model configurations into the task–channel heatmap, 9 of 11 tasks (all except topological sort and coin change) exhibit at least one channel with shifts above 45 pp (Figure 2). Figure 3 illustrates these shifts as distribution changes. Rate limiting (63.4%) and expression parsing (71.5%) have the lowest pass rates, while tree traversal, shortest paths, and topological sort achieve pass rates above 99%. Representative examples:

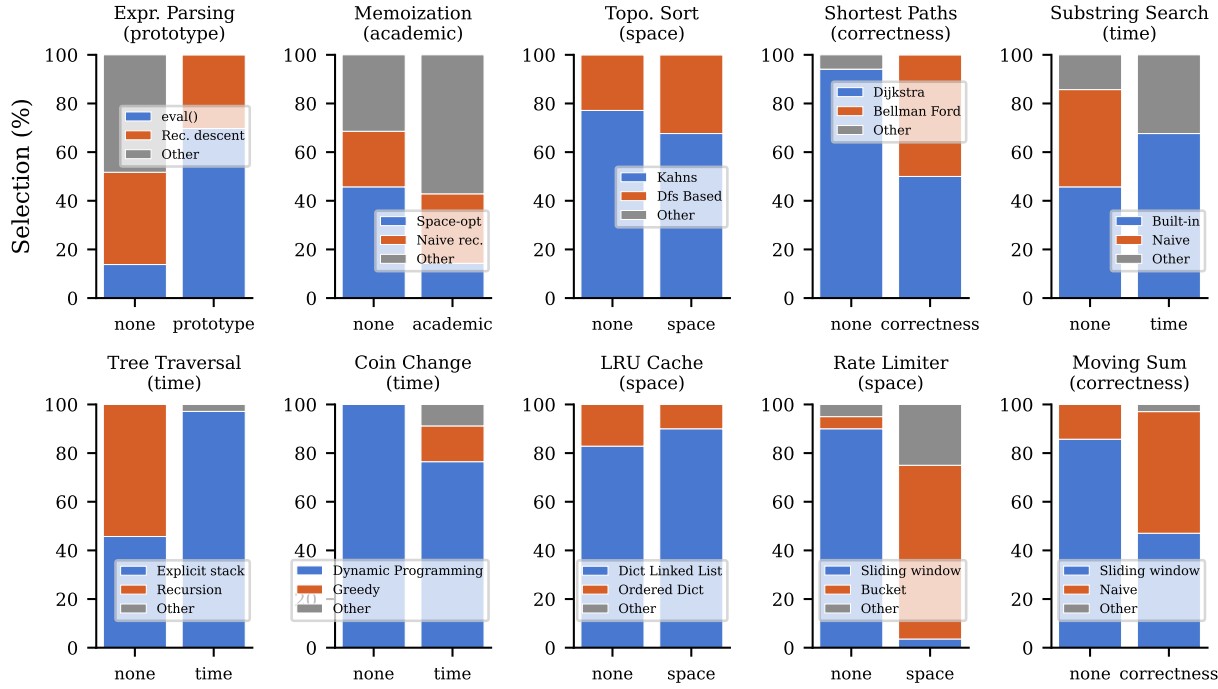

*Figure 3.* Representative steering examples showing algorithm-family distributions before (baseline) and after a cue. Appendix B.2 gives the full per-cue distributions.

- Persona cues select algorithm sophistication: academic persona yields 43% matrix exponentiation vs. 0% under junior, with a corresponding pass rate penalty (80% vs. 100%; Appendix Table 11).
- Context/import cues control shortcut activation and memoization strategy: `prototype` yields 70% `eval` in expression parsing vs. 6% under `interview` (64 pp swing; Appendix Table 10); `import functools` yields 83% top-down memoization vs. 17% without import context (Appendix B).
- Constraint cues produce large effects: tree traversal recursion ranges from 0% (`space`) to 89% (`readability`), an 89 pp swing; explicit stack reaches 97% under `time` (Appendix Table 12). Substring search eliminates naive under `time` (89 pp) and activates KMP (21%; Appendix Table 12).

**Finding.** Within-channel cue changes shift algorithm distributions, and the strongest channel varies by task (RQ1–RQ2), indicating steering is large but task-specific.

### 4.2. Applied Tasks: LRU, Rate Limiting, Moving Sum

We test whether steering effects transfer beyond classical algorithms to applied tasks common in production systems and data pipelines. We run the same cue suite on LRU caching, rate limiting, and moving-sum aggregation; Appendix Table 12 reports per-cue distributions. LRU achieves 99.8% pass, while rate limiting is the hardest applied task

(63.4% pass on the full main grid). Under a space cue, sliding-window use in rate limiting collapses from 90.0% (no constraint cue) to 3.6% and shifts toward bucketed counters (71.4%). In moving sum, a correctness cue shifts the model from a nearly pure sliding-window baseline (85.7%) to a near-even split between sliding-window and naive implementations (47.1% vs. 50.0%).

**Finding.** Steering transfers to applied tasks with large deltas, indicating that prompt context affects production-style algorithm choices as well as textbook problems.

### 4.3. Mechanism Ablation: Semantic vs. Surface Cues

To distinguish semantic from surface effects, we apply paired memoization cues with matched meaning but different surface form (e.g., `time-critical` vs. `TIME-CRITICAL!!`). All time/performance phrasings increase matrix exponentiation from 0% baseline to 14–41% (Appendix Table 13).

**Finding.** Meaning drives steering; surface form adds negligible additional effect once semantic content is held constant.

### 4.4. Model Comparison

RQ3 asks whether steering persists across models and temperatures. Across 15 configurations (Appendix Tables 9 and 14), pass rates span 72.0% (Devstral) to 96.1% (Claude Sonnet 4); 5.2% of outputs are unknown overall; pass rates

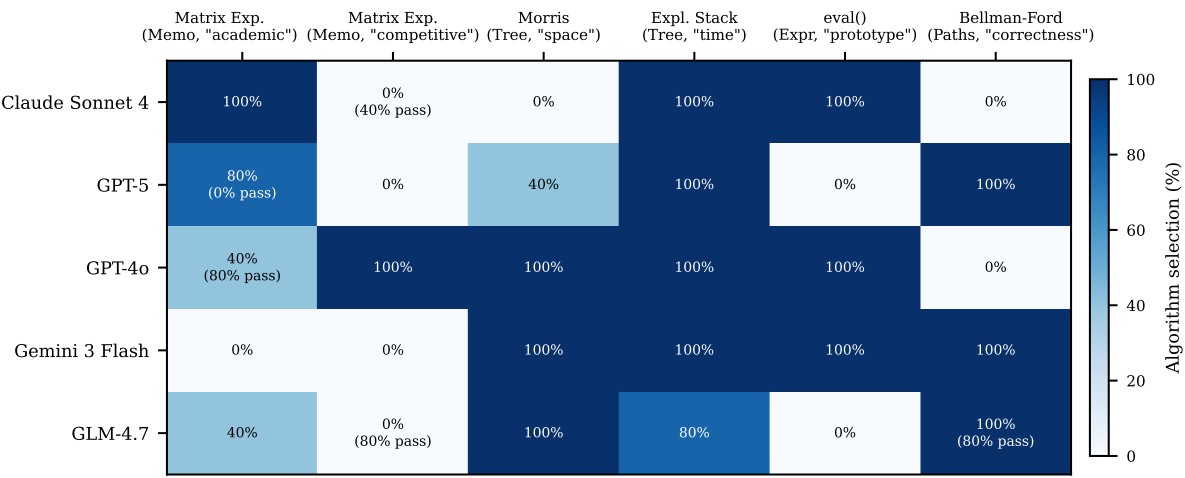

*Figure 4.* Model divergence under identical cues. Each cell shows a target algorithm's selection rate for one model–cue pair; pass rate is annotated when below 85%. Columns span three tasks and six cue conditions.

stay high across $T$ while the unknown rate rises slightly. Figure 4 shows the same cue flipping algorithm choice in opposite directions across models.

**Finding.** Steering generalizes across models and tested temperatures; direction varies by model.

**Quantization and Reasoning** Quantization modestly attenuates steering: DeepSeek-R1 70B Llama-distill drops by 1.6 pp in pass rate and 4.5 pp in capacity from FP16 to Q8_0, decoupling steerability from correctness and suggesting model-side interventions warrant study; Qwen2.5-Coder 32B Instruct shows smaller shifts. The reasoning model DeepSeek-R1 70B stays at 89.3% pass with below-mean steering (45.5 pp mean capacity, vs. 50.0 pp API-mean); Appendix C reports quantization and model-diagnostic details.

### 4.5. Expression Parsing: Shortcut Activation

`prototype` activates `eval` in 69.7% of outputs vs. 6.1% under `interview` (64 pp); this task has one of the lowest pass rates (71.5%), driven by brittle `eval` failures (Appendix Table 10; Appendix Figure 6).

**Finding.** Context can trigger unsafe shortcuts, so evaluation must track algorithm families alongside correctness.

### 4.6. Memoization: Persona and Import Effects

We test whether steering toward sophisticated algorithms harms correctness (RQ4) and whether import context steers memoization style. Appendix Table 11 and Appendix Figure 7 report distributions and the tradeoff. Across all 15 model configurations at $T{=}0$, the `academic` persona produces matrix exponentiation in 42.9% of outputs at 80.0% pass, while `junior` and `senior_engineer` stay at 100% pass with space-optimized solutions; im-

port `functools` drives 82.9% top-down memoization vs. 17.1% without import.

**Finding.** More sophisticated algorithm choices can reduce correctness on tasks where they introduce new failure modes (RQ4); see Section 5 for scope, and import context is a powerful steering channel.

### 4.7. Other Tasks

We test whether steering appears broadly across graph and search tasks beyond the case studies above. Constraint cues in topological sort, shortest paths, substring search, and tree traversal yield large, task-specific swings (29.5 pp DFS shift, 44.3 pp Dijkstra swing, 88.6 pp naive swing, 88.6 pp recursion swing), as summarized in Appendix Table 12.

**Finding.** Constraint cues reliably steer algorithm choice across diverse tasks, reinforcing that steering is systematic and task-general.

### 4.8. Transfer to HumanEval: Pass@$k$ Is Blind

To show the gap on a standard benchmark, we ran the cue framework on the subset of HumanEval (Chen et al., 2021) tasks admitting multiple valid algorithms: 4 tasks (including /96 `count_up_to` and /120 top-$k$) $\times$ 3 cues (performance, readability, none) $\times$ 3 API models (Claude, GPT-5, GPT-4o) $\times$ 5 reps = 180 runs (92.2% pass). Steering appears in 5/12 (task, model) pairs. On HumanEval/96, Claude selects Sieve of Eratosthenes 5/5 under a performance cue and trial division 5/5 under a readability cue (pass@1 = 1.0 in both); HumanEval/120 flips between `heapq.nlargest` and a sorted-list approach while correctness remains unchanged.

**Finding.** Cue-driven algorithm shifts persist on standard benchmarks and are invisible to pass@$k$.

# 5. Analysis

## 5.1. Steering Mechanisms

Why do contextual cues influence algorithm selection? We frame three *interpretive hypotheses*: **retrieval by analogy** (cues activate regions associated with particular domains) (Olsson et al., 2022); **constraint propagation** (explicit time/space mentions bias subsequent choices); and **style anchoring** (surface style primes correlated algorithm families). Our semantic-vs-surface ablation on memoization (Section 4.3) is consistent with the first two but does not separate them. Appendix C.2 reports a token log-probability diagnostic as a below-top-1 signal complementing the output-level claims without identifying a mechanism. We leave causal validation (e.g., activation patching, causal mediation) to future work.

## 5.2. Effect Sizes

Maximum steering deltas range from ∼44 to 100 pp, with peaks in tree traversal and substring search under constraint cues, memoization under persona cues, and applied tasks under style/import/context cues. The dominant channel varies by task (e.g., style for expression parsing and LRU, import for rate limiting, persona for memoization), indicating steering is a task-specific mechanism.

## 5.3. Cross-Task Patterns

**Algorithmic Coherence** Steering directions are interpretable: models associate cues with algorithm properties. A `time` constraint activates faster algorithms (KMP over naive search, explicit stack over recursion); a `space` constraint activates memory-efficient approaches (Morris traversal, in-place algorithms). Appendix C.3 reports the one borderline incoherent case we find.

**Sophistication–Reliability Tradeoff (RQ4)** Implicit cues that push the model toward more sophisticated algorithms tend to reduce pass rates, with a task-dependent relationship. On memoization, aggregated across all 15 model configurations, the academic persona produces matrix exponentiation in 42.9% of outputs at 80.0% pass, the competitive persona at 31.4%/85.7%, and the junior persona at 0%/100%. On tree traversal, an 89 pp steering effect across constraint cues preserves correctness. RQ4 is therefore best read as a regional question: which (task, cue) regions exhibit a sophistication–reliability penalty. The appendix scatter (Appendix Figure 7) reports the per-cell distribution and the aggregate $r$ as one summary among many.

**Expert Personas as Model-Specific Risk** Where the previous paragraph identifies *when* the tradeoff appears, this one identifies *which prompts trigger it*: expert-style persona prompts (OpenAI, 2026) shift algorithm choice toward more sophisticated solutions, and on *some* (model, task) pairs this reduces reliability. On memoization, the junior persona reaches 100% pass with simple implementations, while the academic persona drops to 80% pass while attempting sophisticated algorithms 43% of the time. The effect varies by model: Claude under the academic persona produces matrix exponentiation correctly 100% of the time. The evidence rests on memoization, where sophisticated algorithms introduce new failure modes; we treat "expert persona" as a model- and task-specific risk factor rather than a universal anti-pattern. When reliability matters, name the algorithm directly (Section 5.6) and avoid implicit persona cues.

**Model-Specific Responses** Steering varies by model: the same cue can push different models in opposite directions. Under the `academic` persona on memoization, Claude selects matrix exponentiation (100% pass), GPT-5 attempts it and fails (0% pass), while Gemini defaults to naive recursion with 100% pass. Similar divergences appear for other cues (e.g., Morris traversal under space constraints; `eval` usage in parsing), indicating model-specific priors (Chiang et al., 2024). Full model–cue–task tables are in Appendix C.4.

## 5.4. Implications

- **Reproducibility** (Pineau et al., 2021). The same specification can yield different algorithms depending on context, changing performance and maintenance.
- **Evaluation and safety**. Correctness-only benchmarks hide policy shifts (e.g., $O(n^2)$ vs. $O(n \log n)$); neutral context can activate shortcuts (`eval`) or steer toward deprecated/vulnerable patterns (Wang et al., 2025; Pearce et al., 2022).

In AI-assisted workflows that treat generated code as a black box, unit tests confirm correctness while rarely surfacing $O(n^2)$ memory usage where $O(1)$ was possible, or recursive implementations that will overflow the stack on large inputs.

## 5.5. Mitigation and Practical Recommendations

We recommend auditing algorithm choice (complexity class, data structures, recursion depth), standardizing prompts to minimize incidental cues, and adding a lightweight guardrail stress test (e.g., Fibonacci $n=36$ or deep recursion) to surface algorithmic fragility. Beyond these post-hoc audits, explicit algorithm specification is the most reliable proactive mitigation in our experiments.

## 5.6. Mitigation: Explicit Algorithm Specification

The most reliable mitigation we tested is direct algorithm naming. We ran 120 controlled experiments (3 tasks × 2–3 target algorithms × 3 API models × 5 reps) in which the

prompt explicitly requests the named algorithm. Pass rate is 120/120 (100%); strict compliance is 73/120 (60.8%) and 118/120 (98.3%) after normalizing for label-granularity synonyms documented in Appendix A.6 (e.g., "dynamic programming" satisfied by bottom-up tabulation). The two genuine non-compliances are both GPT-5 producing an iterative stack instead of Morris traversal (code still passes), suggesting Morris is at the edge of GPT-5's repertoire.

Explicit specification changes the outcome. Under the `academic` persona on memoization, GPT-5 attempts matrix exponentiation in 4/5 runs and passes 0/5; explicitly asking GPT-5 for matrix exponentiation gives 5/5 compliance and 5/5 pass. Same model, same algorithm, different prompt.

High explicit-instruction compliance and high implicit-context steering coexist: 8 of the 9 tested (task, model) pairs reach $\geq$90% lenient compliance, yet the same three models exhibit median 40 pp implicit-context steering across 556 task–channel cells on the main grid. Compliance with direct instructions does not protect against incidental-cue drift.

**Practitioner guidelines**    Because correctness alone cannot expose these shifts, we translate the measurements into four operational guidelines:

1. Specify the algorithm directly when its choice has performance, security, or maintenance implications; avoid persona/role/constraint cues as substitutes.
2. When direct specification is impractical, constraint cues more often dominate persona cues when both are present: across 7 tested (persona, constraint) conflict cells (memoization, tree traversal, rate limiter), the combined cue matches the constraint dominant in 4/7 (persona in 1/7).
3. Standardize system prompts, surrounding code, project metadata, and imports across runs to make algorithm choice reproducible.
4. Test across models: steering direction varies (Section 5); a cue harmless on one can flip choice on another.

**Power and noise floor**    Per-cell power at $\alpha$=0.05 for a 15 pp shift is 25–44%; the design is underpowered near the threshold but well above it for the 40–90 pp shifts we report (Appendix A.7). Classifier noise is $\approx \pm 5$ pp, well below the 40 pp median.

### 5.7. Limitations

- **Task coverage**: 8 classical + 3 applied tasks; broader domains (web frameworks, multi-file) remain open.
- **Model coverage**: 15 configurations including 5 API models; behavior may shift with model updates.
- **Cues**: Designed to be clear and isolated; real prompts are noisier and multi-cue.
- **Classifier**: 87–88% agreement and substantial unknown rates; measurement error can bias effect sizes.

- **Tests**: Unit tests check correctness; scalability requires separate stress tests (we add Fibonacci $n$=36 and deep-recursion guardrails).

## 6. Related Work

**Prompt Sensitivity and Prompt Engineering**    Prior work shows that LLM outputs vary with prompt formulation. Liu et al. (2024) document positional sensitivity, and Sclar et al. (2024) quantify sensitivity to spurious formatting. Researchers engineer prompts to optimize outcomes, including misleading context (Lam et al., 2025), anchoring (Tian & Zhang, 2025), secure prompting (Tony et al., 2025), and temporal markers that steer models toward deprecated APIs (Wang et al., 2025). These works focus on success/failure. We measure *which correct algorithm* is chosen and show that neutral context also steers choice.

**Code Generation Evaluation**    Standard benchmarks emphasize functional correctness. HumanEval (Chen et al., 2021) and MBPP (Austin et al., 2021) use pass@$k$, treating all correct outputs as equivalent; surveys call for richer evaluation (Liu et al., 2023). We show that correctness hides algorithm-policy shifts (up to 100 pp) driven by context.

**Instruction Hierarchy, Reasoning, and Personas**    Instruction hierarchy work shows that system-level cues can override user intent (Wallace et al., 2024). Reasoning and persona prompts have mixed accuracy effects (Kojima et al., 2022; Yao et al., 2023; Zheng et al., 2024). We connect these strands by showing that persona and metadata cues shift algorithm selection, leave correctness unimproved, and can reduce reliability (OpenAI, 2026).

**Algorithmic Diversity and Developer Interaction**    Lee et al. (2025) study intrinsic diversity by sampling many outputs under fixed prompts; we study contextual steering by holding sampling fixed and varying context. The two are complementary: their method reveals the repertoire; ours shows how context selects within it. Developers often accept suggestions with limited review (Vaithilingam et al., 2022; Barke et al., 2023; Mozannar et al., 2024), amplifying the cost of invisible selection.

## 7. Conclusion

Across 46,535 experiments, prompt context steers algorithm selection by up to 100 pp—reliability tradeoffs invisible to correctness-only evaluation. Alongside pass@$k$, evaluations should report algorithm-family distributions and channel-level max–min capacity (Section 2). Released classifiers and data: `https://github.com/mpi-dsg/invisible-lottery`.

## Impact Statement

Cue-induced algorithm steering already happens whenever developers use LLM-based coding assistants; without measurement, it remains invisible. This work documents the phenomenon so that developers, evaluators, and model providers can act on it by auditing algorithm choice, standardizing incidental context, and looking past pass-rate correctness. The same findings could in principle be used to nudge models toward insecure or inefficient algorithms, but that risk exists today undiagnosed; making the channel visible lets defenders, reviewers, and downstream evaluations respond.

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

# Appendix

## A. Experimental Setup and Measurement Checks

This appendix supports Sections 3 and 3.5 by giving the full cue/task definitions, prompt structure, classifier details, and measurement checks used to interpret the main results.

### A.1. Cue Type Taxonomy

This subsection expands the steering-channel taxonomy from Section 2 with the exact cue values used in the experiments.

### A.2. Task Specifications

This subsection expands the task suite from Section 3 with full signatures and the valid algorithm families used as classifier labels.

### A.3. Prompt Templates

This subsection records the prompt structure used in Section 3; each generated prompt combines a base task specification, one cue injection, examples, and a required function signature. Representative cue injections include:

- **Space constraint**: "Memory Critical: This runs on embedded devices with very limited RAM. Minimize memory usage."
- **Time constraint**: "Performance Critical: This function is called millions of times per day. Execution speed is the top priority."
- **Academic persona** (system): "You are a computer science professor specializing in algorithm analysis. You prefer algorithms with proven bounds."
- **Junior persona** (system): "You are a junior developer who recently completed a coding bootcamp. You prefer simple, straightforward approaches."

### A.4. Algorithm Detection Methodology

This subsection expands Section 3 with the structural signals used to classify generated code into algorithm families. For each task, we define canonical patterns:

- **Control flow**: Loop types (for/while), recursion presence, nesting depth
- **Data structures**: Stack/queue usage, dictionary patterns, set operations
- **Decorators**: `@lru_cache`, `@cache`, custom memoization
- **Imports**: `functools`, `collections`, `heapq`
- **Builtins**: `eval`, `literal_eval`, `sorted`, `set`
- **Applied-task patterns**: sliding-window deque usage, prefix-sum arrays, and bucketed counters

Each pattern contributes to a confidence score; we mark classifications with confidence below 0.3 as `unknown`. The framework explicitly detects shortcut families (e.g., `eval` for expression parsing) as distinct from proper implementations.

### A.5. Threshold Sensitivity

This subsection checks whether the 0.3 classifier confidence threshold used in Section 3 drives the reported steering effects. Raising the threshold increases the unknown rate, but the largest effects remain stable except where hybrid implementations blur family boundaries.

### A.6. Classifier Calibration: LLM-as-Judge

This subsection reports the calibration check referenced in Section 3. We ran GPT-4o as an LLM judge on 110 stratified samples and compared judge labels against AST classifier labels. Of 110 samples, 80 were exact matches (73%), 17 were label-granularity differences where both labels are correct at different specificity levels (e.g., `dynamic_programming` vs. `bottom_up_tabulation`), 8 were ambiguous edge cases, and 5 were genuine disagreements where one label was clearly preferable ($\sim$5% genuine error rate). After normalizing for label granularity, agreement is 88%.

We use this as a noise-floor estimate. Judge superiority over the AST classifier is outside this calibration claim. Treating the 13% AST disagreement rate from the main-text validation as the conservative noise floor, $\text{SE} = \sqrt{p(1-p)/N}$ gives approximately $\pm 5$ pp at $N{=}40$ (the high-replication robustness sample) and $\pm 5.7$ pp at the typical main-grid per-cell $N{=}35$, both well below the 40 pp median effect.

### A.7. Power Analysis

This subsection justifies the $\epsilon{=}15$ pp steering threshold defined in Section 2. The main-grid per-cell $N$ is 35 (5 API $\times$ 5 reps + 8 local $\times$ 1 + 2 sweep $\times$ 1); the high-replication study uses $N{=}40$ (2 models $\times$ 20 reps). Approximate per-cell power for detecting a 15 pp shift with a two-sided $\alpha{=}0.05$ two-proportion test (Cohen's $h$) is 24.7% at $N{=}35$ and 27.9% at $N{=}40$ when $p_1{=}0.50$, and 39.3%/43.7% when $p_1 \in \{0.10, 0.90\}$. The design is therefore underpowered for minimal 15 pp shifts, so null results near that threshold should be read conservatively; this limitation falls short of explaining the repeated 40–90 pp effects reported in the main text.

### A.8. Temperature Persistence

This subsection supports the temperature discussion in Section 2 by repeating the cue framework on two local models

| Category | Channel type | Cue values (all used) |
|---|---|---|
| Persona | Persona | academic, competitive, junior, none, senior_engineer |
| Context | Context | interview, none, production, prototype, teaching |
| | Import | collections, functools, heapq, itertools, minimal, none |
| | Dependency | data_heavy, none, scientific, stdlib_only, web |
| | Benchmark | batch_processing, high_volume, latency_sensitive, memory_constrained, none |
| | Code Review | conciseness, defensive, none, performance, readability |
| Constraint | Constraint | correctness, none, readability, space, time |
| | Test Complexity | adversarial, edge, none, scale, trivial |
| Style | Style | legacy, modern, none, terse, verbose |
| | Docstring Style | google, minimal, none, numpy, sphinx |
| | Comment Density | heavy, moderate, none, sparse |
| | Variable Naming | abbreviated, descriptive, none, single_letter |
| | Error Handling | eafp, lbyl, minimal, none |
| Format | Input Format | doctest, json, plain, table |
| Temporal | Temporal | 2008, 2014, 2020, 2024, none |
| Control | Placebo | id_12857, id_58291, id_73046, none, ref_91c5a0, ref_a7b3c9, ref_d4e8f2, ver_001, ver_002, ver_003 |
| | Innocuous | color_blue, color_green, color_orange, none, project_alpha, project_beta, project_gamma, team_dragon, team_phoenix |
| | Interaction | academic_space, academic_time, competitive_time, interview_time, junior_readability, junior_space, junior_time, legacy_senior, modern_junior, none, production_correctness, production_time, prototype_readability, senior_space, senior_time |
| Ablation (memoization-only) | Ablation | none, performance_sensitive, time_critical_caps, time_critical_code, time_critical_natural |

*Table 3.* Complete steering-channel taxonomy used to construct prompts. Rows list each channel type and the exact cue values evaluated; memoization-only ablation cues are excluded from Figure 2.

with user-controllable temperature.

### A.9. High-Replication Robustness

This subsection checks whether five-replication estimates are artifacts of low replication. We re-ran six representative cells at 20 replications each across two API models, for 240 runs total.

### A.10. Failure Mode and Guardrail Checks

This subsection expands the non-functional guardrail discussion from Section 3. Of memoization outputs that attempted matrix exponentiation and then failed functional tests (17 cases), 15 (88.2%) are base-case errors at $n \leq 2$ and 2 (11.8%) are logic bugs at intermediate $n$; large-$n$ failures do not appear in this breakdown. Per-stress-test family outcomes follow.

| Task | Signature | Algorithm Families |
|------|-----------|--------------------|
| Expression Parsing | `evaluate(expr: str) -> int` | Recursive descent, shunting-yard, precedence climbing, `eval` shortcut |
| Memoization | `fibonacci(n: int) -> int` | Naive recursion $O(2^n)$, memoized $O(n)$, bottom-up DP, space-optimized $O(1)$, matrix exp. $O(\log n)$ |
| Topological Sort | `topological_sort(graph) -> list` | DFS-based (post-order), Kahn's algorithm (BFS) |
| Shortest Paths | `shortest_path(graph, start, end)` | Dijkstra $O((V+E)\log V)$, Bellman–Ford $O(VE)$, BFS (unit-weight) |
| Substring Search | `find_substring(text, pattern)` | Naive $O(nm)$, KMP $O(n+m)$, Boyer–Moore, Rabin–Karp |
| Tree Traversal | `inorder(root) -> list` | Recursive, iterative stack, Morris traversal $O(1)$ space |
| Deduplication | `deduplicate(items) -> list` | Hash set (order-preserving), `dict.fromkeys` |
| Coin Change | `coin_change(coins, amount)` | Bottom-up DP, top-down DP (memoized), BFS |
| LRU Cache | `LRUCache(capacity)` class | Dict + doubly-linked list, `OrderedDict`, `functools.lru_cache` |
| Rate Limiter | `allow_requests(timestamps, window_s, max_requests)` | Naive scan, sliding-window deque, bucketed counter |
| Moving Sum | `moving_sum(values, k)` | Naive recompute, prefix sum, sliding window |

*Table 4.* Task signatures and valid algorithm families for the 11-task benchmark. These families define the classifier labels and result distributions used throughout the paper; task constraints make each listed family functionally valid for the corresponding tests.

| Condition | $\tau{=}0.3$ | $\tau{=}0.5$ | $\tau{=}0.7$ |
|-----------|---------|---------|---------|
| Tree traversal + constraint | 88.6 | 88.6 | 88.6 |
| Memoization + persona | 85.7 | 87.5 | 87.5 |
| Expression parsing + context | 65.7 | 65.7 | 62.8 |
| Unknown rate | 5.2% | 9.4% | 22.9% |

*Table 5.* Classifier-threshold sensitivity at $T{=}0$ for three representative high-effect cells. Entries report max–min algorithm-family steering deltas in pp (best family per cell, classified denominator) under confidence thresholds 0.3, 0.5, and 0.7; the final row reports the overall unknown-label rate over the $T{=}0$ main grid (39,696 runs). The three cell-level deltas are essentially invariant to the threshold; raising $\tau$ mostly reclassifies low-confidence outputs as unknown without disturbing the dominant family.

| Model / Metric | $T{=}0.0$ | $T{=}0.3$ | $T{=}0.7$ | $T{=}1.0$ |
|----------------|-------|-------|-------|-------|
| Qwen2.5-Coder 32B | | | | |
| mean capacity (pp) | 49.7 | 55.1 | 59.4 | 64.7 |
| pass rate | 93.0% | 92.8% | 92.2% | 91.4% |
| DeepSeek-R1 70B | | | | |
| mean capacity (pp) | 45.5 | 44.9 | 50.3 | 45.8 |
| pass rate | 89.3% | 87.2% | 87.7% | 86.3% |

*Table 6.* Temperature robustness for Qwen2.5-Coder 32B and DeepSeek-R1 70B (base, $N{=}1{,}135$ per cell). For each temperature, the table reports mean channel-level steering capacity (max–min pp across cue values, averaged over (task, cue-type) cells with at least two cue values, classified denominator, `input_format` excluded for consistency with Table 2) and pass rate. Steering capacity remains non-trivial at every temperature while pass rates stay at 86–93%, indicating the cue effect is not a $T{=}0$ artifact.

| Task | Cue | Claude (20 reps) | GPT-4o (20 reps) |
|---|---|---|---|
| Memoization | academic | matrix_exp 20/20 | space_opt 20/20 |
| Memoization | junior | space_opt 20/20 | space_opt 20/20 |
| Memoization | none | space_opt 20/20 | space_opt 20/20 |
| Tree traversal | space | Morris 20/20 | Morris 20/20 |
| Tree traversal | readability | recursion 20/20 | recursion 20/20 |
| Tree traversal | none | recursion 20/20 | recursion 20/20 |

*Table 7.* High-replication robustness for six representative (task, cue) cells. Claude Sonnet 4 and GPT-4o were each run 20 times per cell; all twelve model–cell combinations are unanimous, so the five-replication estimates used in the main analysis are stable in these checked cases. The GPT-4o + memoization + `academic` cell shows space_opt 20/20 here but matrix_exp 2/5, bottom_up_-tabulation 2/5, naive_recursion 1/5 (4/5 pass; 0 unknowns) in the main grid; this may reflect model-version drift between the main-grid run and this robustness run (a known constraint of remote API evaluation).

| Guardrail | Algorithm | Pass | Total | Fail% |
|---|---|---|---|---|
| Fibonacci $n=36$ | naive recursion | 407 | 443 | 8.1% |
| Fibonacci $n=36$ | bottom-up DP | 73 | 77 | 5.2% |
| Fibonacci $n=36$ | matrix exp | 153 | 166 | 7.8% |
| Fibonacci $n=36$ | space-opt DP | 3,209 | 3,362 | 4.6% |
| Fibonacci $n=36$ | top-down memo | 133 | 265 | 49.8% |
| Tree depth 2000 | recursion | 1,633 | 2,332 | 30.0% |
| Tree depth 2000 | explicit stack | 1,386 | 1,711 | 19.0% |
| Tree depth 2000 | Morris | 56 | 70 | 20.0% |
| Tree depth 2000 | generator | 2 | 10 | 80.0% |

*Table 8.* Guardrail stress-test outcomes by algorithm family for the non-functional checks described in Section 3. Rows report pass counts, totals, and failure rates for Fibonacci $n=36$ with a 1-second timeout (computed over all memoization-task outputs) and tree depth 2000 (computed over all tree-traversal outputs) across all 15 model configurations and all temperatures, walking the canonical results directory and excluding `_extra_runs`. Combined, naive recursion on Fibonacci plus recursion on tree traversal fail 735 of 2,775 stress invocations (26.5%), confirming that recursive families incur real performance costs that the non-recursive variants avoid.

# B. Extended Experimental Results

This appendix extends Section 4 with figures and tables that support the main experimental claims without repeating their interpretation.

## B.1. Steering Figures

This subsection collects appendix figures referenced from the main results and analysis sections.

## B.2. Full Result Tables

This subsection provides the per-model, per-cue, and ablation tables referenced from Section 4.

# C. Additional Model and Diagnostic Analyses

This appendix collects secondary analyses referenced from Section 5; each item adds a specific diagnostic without restating the main result.

## C.1. Quantization and Reasoning Models

This subsection expands the model-comparison discussion with quantized model pairs and the DeepSeek-R1 reasoning model.

## C.2. Token Log-Probability Diagnostic

This diagnostic supports the mechanism discussion by checking whether a cue can move alternatives into the token candidate set while the realized top-1 algorithm remains fixed. We collected top-10 token log-probabilities on GPT-4o for memoization under three persona conditions (academic, junior, none) at five replications each. The realized algorithm family was `space_optimized` in 15/15 generations. Tokens associated with matrix exponentiation ("matrix", "exponent", etc.) entered the top-10 candidate set in all five academic reps, with per-rep peak probabilities of 76%, 95%, 99.9%, 99.9%, 99.9% at decision-relevant positions. Under junior, the same tokens appeared in 1/5 reps with a peak probability of 1.2%; under none, they appeared in 5/5 reps as low-probability candidates (peaks 0–7.6%). This is a complementary signal only; the paper's central claim is the output-level distribution shift.

## C.3. Incoherent Steering Check

This subsection supports the analysis of algorithmic coherence by listing cases where a constraint cue reduces the expected algorithm family rather than activating it. Across the $T=0$ main grid one case meets this criterion: a space cue lowers sliding-window use for moving sum by 5.1 pp relative to the `none` baseline. The remaining expected mappings all move in the predicted direction (e.g., the time cue raises Dijkstra use slightly, $+0.2$ pp, on shortest paths).

## C.4. Model Divergence Tables

This subsection expands the model-specific response discussion with top divergence cases across API models.

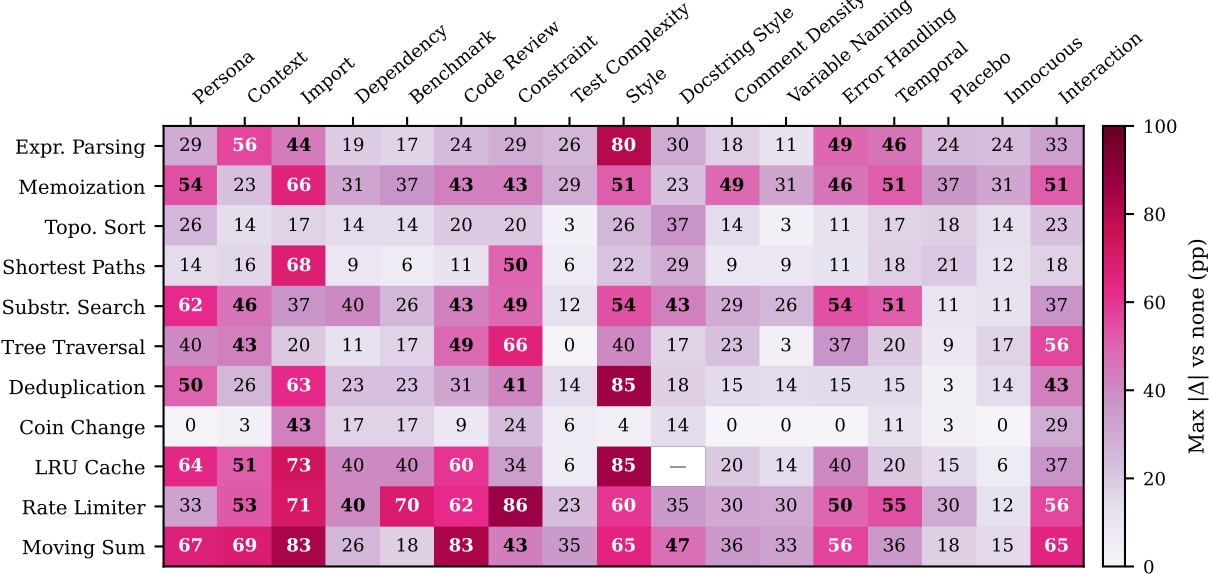

| | Persona | Context | Import | Dependency | Benchmark | Code Review | Constraint | Test Complexity | Style | Docstring Style | Comment Density | Variable Naming | Error Handling | Temporal | Placebo | Innocuous | Interaction |
|---|---|---|---|---|---|---|---|---|---|---|---|---|---|---|---|---|---|
| Expr. Parsing | 29 | **56** | **44** | 19 | 17 | 24 | 29 | 26 | **80** | 30 | 18 | 11 | **49** | **46** | 24 | 24 | 33 |
| Memoization | **54** | 23 | **66** | 31 | 37 | **43** | **43** | 29 | **51** | 23 | **49** | 31 | **46** | **51** | 37 | 31 | **51** |
| Topo. Sort | 26 | 14 | 17 | 14 | 14 | 20 | 20 | 3 | 26 | 37 | 14 | 3 | 11 | 17 | 18 | 14 | 23 |
| Shortest Paths | 14 | 16 | **68** | 9 | 6 | 11 | **50** | 6 | 22 | 29 | 9 | 9 | 11 | 18 | 21 | 12 | 18 |
| Substr. Search | **62** | **46** | 37 | 40 | 26 | **43** | **49** | 12 | **54** | **43** | 29 | 26 | **54** | **51** | 11 | 11 | 37 |
| Tree Traversal | 40 | **43** | 20 | 11 | 17 | **49** | **66** | 0 | 40 | 17 | 23 | 3 | 37 | 20 | 9 | 17 | **56** |
| Deduplication | **50** | 26 | **63** | 23 | 23 | 31 | **41** | 14 | **85** | 18 | 15 | 14 | 15 | 15 | 3 | 14 | **43** |
| Coin Change | 0 | 3 | **43** | 17 | 17 | 9 | 24 | 6 | 4 | 14 | 0 | 0 | 0 | 11 | 3 | 0 | 29 |
| LRU Cache | **64** | **51** | **73** | 40 | 40 | **60** | 34 | 6 | **85** | — | 20 | 14 | 40 | 20 | 15 | 6 | 37 |
| Rate Limiter | 33 | **53** | **71** | **40** | **70** | **62** | **86** | 23 | **60** | 35 | 30 | 30 | **50** | **55** | 30 | 12 | **56** |
| Moving Sum | **67** | **69** | **83** | 26 | 18 | **83** | **43** | 35 | **65** | **47** | 36 | 33 | **56** | 36 | 18 | 15 | **65** |

Max |Δ| vs none (pp)

*Figure 5.* Baseline-referenced steering across all task–channel pairs. Each cell reports the largest |Δ| in pp relative to the `none` baseline across cue values within the channel; Figure 2 reports the max–min version. The same task-specific peaks appear under both views, with constraint cues strongest for substring search and tree traversal.

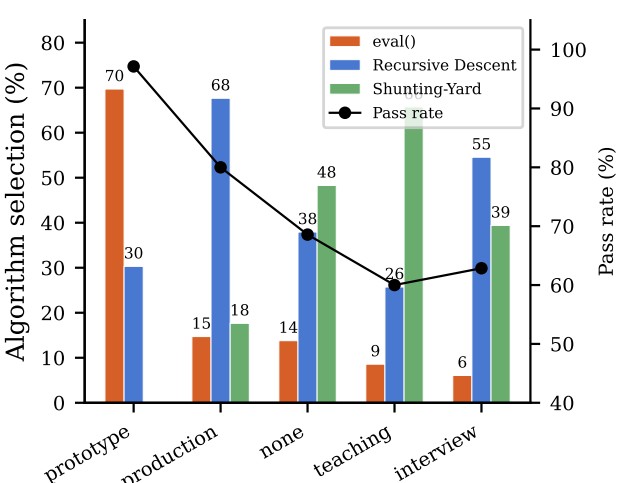

*Figure 6.* Expression-parsing algorithm distribution by context cue. Bars report selection rates for `eval`, recursive descent, and shunting-yard; the black line reports pass rate on the right axis. Holding the task specification fixed, context shifts `eval` usage from 70% under `prototype` to 6% under `interview`.

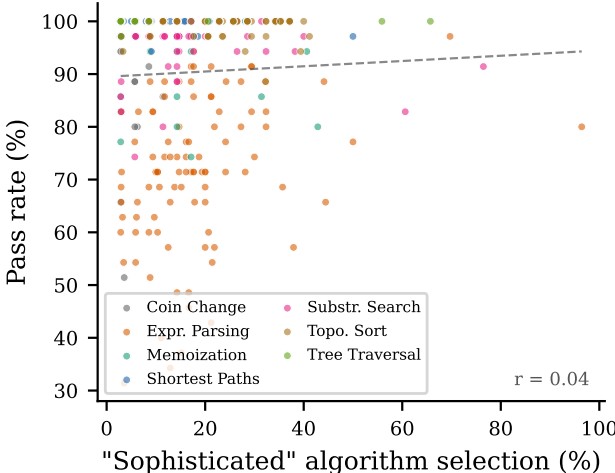

*Figure 7.* Sophistication–reliability diagnostic across task–cue conditions. Each point is one condition; the x-axis is the share of outputs selecting a task-specific sophisticated family (e.g., matrix exponentiation for memoization, `eval` for expression parsing), and the y-axis is pass rate. The aggregate Pearson correlation is near zero ($r \approx 0.09$), masking task-specific tradeoffs (memoization with academic personas reduces reliability; tree traversal under constraint cues does not); Section 5 interprets at the task and cue level.

| Model Configuration | N | Pass | Pass% | Unknown% |
|---|---|---|---|---|
| *API Models (T=0.0)* | | | | |
| GPT-5 | 5,667 | 5,330 | 94.1% | 4.6% |
| GPT-4o | 5,667 | 5,395 | 95.2% | 3.1% |
| Gemini 3 Flash Preview | 5,669 | 5,410 | 95.4% | 6.1% |
| GLM-4.7 | 5,675 | 5,016 | 88.4% | 5.1% |
| Claude Sonnet 4 | 5,668 | 5,446 | 96.1% | 6.3% |
| *Local Models, single T (T=0.0)* | | | | |
| DeepSeek-Coder v2 16B | 1,135 | 976 | 86.0% | 4.8% |
| Llama 3.3 70B | 1,135 | 983 | 86.6% | 6.0% |
| Devstral | 1,135 | 817 | 72.0% | 20.7% |
| Qwen2.5-Coder 7B | 1,135 | 955 | 84.1% | 4.9% |
| *Local Models, temperature sweep* | | | | |
| Qwen2.5-Coder 32B (T=0.0) | 1,135 | 1,055 | 93.0% | 1.6% |
| Qwen2.5-Coder 32B (T=0.3) | 1,135 | 1,053 | 92.8% | 1.9% |
| Qwen2.5-Coder 32B (T=0.7) | 1,135 | 1,047 | 92.2% | 2.7% |
| Qwen2.5-Coder 32B (T=1.0) | 1,135 | 1,037 | 91.4% | 3.3% |
| DeepSeek-R1 70B (T=0.0) | 1,135 | 1,013 | 89.3% | 1.4% |
| DeepSeek-R1 70B (T=0.3) | 1,135 | 990 | 87.2% | 7.4% |
| DeepSeek-R1 70B (T=0.7) | 1,135 | 995 | 87.7% | 7.2% |
| DeepSeek-R1 70B (T=1.0) | 1,135 | 979 | 86.3% | 7.7% |
| *Aggregates* | | | | |
| All configurations at $T=0$ (15 configs) | 39,696 | 36,530 | 92.0% | 5.2% |
| **Canonical released grid (all $T$)** | **46,535** | **42,657** | **91.7%** | **5.2%** |

*Table 9.* Pass and unknown-label rates by model configuration on the canonical 46,535-run released grid. Eleven distinct models are shown across seventeen rows: five API models and four single-$T$ local models at $T=0$, plus Qwen2.5-Coder 32B and DeepSeek-R1 70B at each of $T \in \{0.0, 0.3, 0.7, 1.0\}$. Four additional quantized configurations (DeepSeek-R1 70B fp16/q8_0, Qwen2.5-Coder 32B fp16/q8_0; each $N=1,135$ at $T=0$) are reported separately in Table 14; together with the eleven distinct models above they form the 15 model configurations at $T=0$ summed in the "All configurations at $T=0$" row (39,696 runs). Adding the six sweep-temperature rows for Qwen2.5-Coder 32B and DeepSeek-R1 70B plus the 29 API retries at $T > 0$ recovers the 46,535 canonical total. Pass rates stay at or above 72% across all configurations.

| Context | N | Pass% | eval% | RecDesc% | Shunting% |
|---|---|---|---|---|---|
| Prototype | 35 | 97.1% | 69.7% | 30.3% | 0.0% |
| None | 35 | 68.6% | 13.8% | 37.9% | 48.3% |
| Production | 35 | 80.0% | 14.7% | 67.6% | 17.6% |
| Teaching | 35 | 60.0% | 8.6% | 25.7% | 65.7% |
| Interview | 35 | 62.9% | 6.1% | 54.5% | 39.4% |

*Table 10.* Expression-parsing results by context cue, aggregated across all 15 model configurations at $T=0$ ($N=35$ runs per cue from the canonical released grid). Pass% uses the raw $N$ denominator; algorithm-family columns use the classified-output denominator (excluding unknown). Holding the task specification fixed, context shifts `eval` shortcut usage from 69.7% under `Prototype` to 6.1% under `Interview`; pass rates vary from 60.0% to 97.1%. RecDesc is recursive descent; Shunting is shunting-yard.

| Persona | N | Pass | MExp | BUDP | Space | TopDn |
|---|---|---|---|---|---|---|
| Academic | 35 | 80.0% | 42.9% | 14.3% | 14.3% | 0.0% |
| Competitive | 35 | 85.7% | 31.4% | 2.9% | 40.0% | 0.0% |
| Junior | 35 | 100.0% | 0.0% | 0.0% | 100.0% | 0.0% |
| None | 35 | 100.0% | 0.0% | 2.9% | 45.7% | 28.6% |
| Senior Engineer | 35 | 100.0% | 0.0% | 0.0% | 91.4% | 5.7% |

*Table 11.* Memoization results by persona cue, aggregated across all 15 model configurations at $T=0$ ($N=35$ runs per cue from the canonical released grid). Pass% uses the raw $N$ denominator; algorithm-family columns use the classified-output denominator (excluding unknown). Academic and competitive personas activate matrix exponentiation in 42.9% and 31.4% of outputs and have lower pass rates; junior and senior-engineer personas avoid matrix exponentiation and pass all tests. MExp is matrix exponentiation; BUDP is bottom-up tabulation; Space is space-optimized DP; TopDn is top-down memoization.

| Task Constraint | N | Pass% | Algo 1% | Algo 2% | Algo 3% |
|---|---|---|---|---|---|
| *Topological Sort* | | | | | |
| Correctness | 35 | 100.0% | 97.1% (Kahn) | 2.9% (DFS) | – |
| Time | 35 | 94.3% | 96.7% (Kahn) | 3.3% (DFS) | – |
| Readability | 35 | 100.0% | 94.3% (Kahn) | 5.7% (DFS) | – |
| None | 35 | 100.0% | 77.1% (Kahn) | 22.9% (DFS) | – |
| Space | 35 | 97.1% | 67.6% (Kahn) | 32.4% (DFS) | – |
| *Shortest Paths* | | | | | |
| Time | 35 | 100.0% | 94.3% (Dijkstra) | 5.7% (BFS) | – |
| None | 35 | 100.0% | 94.1% (Dijkstra) | 5.9% (BFS) | – |
| Readability | 35 | 100.0% | 80.0% (Dijkstra) | 4.0% (BFS) | 16.0% (B-F) |
| Space | 35 | 97.1% | 70.4% (Dijkstra) | 11.1% (BFS) | 18.5% (B-F) |
| Correctness | 35 | 97.1% | 50.0% (Dijkstra) | – | 50.0% (B-F) |
| *Tree Traversal* | | | | | |
| Readability | 35 | 100.0% | 88.6% (Rec) | 11.4% (Stack) | – |
| None | 35 | 100.0% | 54.3% (Rec) | 45.7% (Stack) | – |
| Correctness | 35 | 100.0% | 31.4% (Rec) | 68.6% (Stack) | – |
| Time | 35 | 100.0% | – | 97.1% (Stack) | 2.9% (Morris) |
| Space | 35 | 100.0% | – | 34.3% (Stack) | 65.7% (Morris) |
| *Substring Search* | | | | | |
| Correctness | 35 | 82.9% | 73.5% (Built-in) | 23.5% (Naive) | 2.9% (KMP) |
| Time | 35 | 94.3% | 67.6% (Built-in) | – | 20.6% (KMP) |
| None | 35 | 97.1% | 45.7% (Built-in) | 40.0% (Naive) | 14.3% (KMP) |
| Readability | 35 | 100.0% | 43.3% (Built-in) | 56.7% (Naive) | – |
| Space | 35 | 100.0% | 5.7% (Built-in) | 88.6% (Naive) | 2.9% (KMP) |
| *Rate Limiter (Applied)* | | | | | |
| None | 22 | 68.2% | 90.0% (Window) | 5.0% (Naive) | 5.0% (Bucket) |
| Correctness | 35 | 57.1% | 64.7% (Window) | 17.6% (Naive) | 17.6% (Bucket) |
| Readability | 35 | 65.7% | 62.5% (Window) | 18.8% (Naive) | 18.8% (Bucket) |
| Time | 19 | 52.6% | 61.5% (Window) | – | 38.5% (Bucket) |
| Space | 35 | 11.4% | 3.6% (Window) | 25.0% (Naive) | 71.4% (Bucket) |
| *Moving Sum (Applied)* | | | | | |
| Time | 35 | 94.3% | 96.9% (Window) | 3.1% (Naive) | – |
| None | 35 | 91.4% | 85.7% (Window) | 14.3% (Naive) | – |
| Space | 35 | 25.7% | 80.6% (Window) | 19.4% (Naive) | – |
| Correctness | 35 | 88.6% | 47.1% (Window) | 50.0% (Naive) | 2.9% (Prefix) |
| Readability | 35 | 91.4% | 42.4% (Window) | 57.6% (Naive) | – |

*Table 12.* Constraint-cue results across six tasks, aggregated across all 15 model configurations at $T{=}0$ from the canonical released grid. Pass% uses the raw $N$ denominator; algorithm-family rates use the classified-output denominator (excluding unknown). Rows within each task block are ordered by Algo 1 share descending. $N{=}35$ in most cells (one cell per model at the listed constraint cue); a few applied-task cells have smaller $N$ because not all models produced runs in that (task, cue) slot. Time and space cues often shift selection without changing the task specification, including stack over recursion for tree traversal and bucket counters for rate limiting.

| Phrasing | N | Pass% | Space-Opt% | Matrix Exp% | Top-Down% |
|---|---|---|---|---|---|
| none (baseline) | 35 | 97.1% | 45.7% | 0.0% | 25.7% |
| Performance Sensitive | 35 | 100.0% | 60.0% | 17.1% | 17.1% |
| Time Critical Caps | 35 | 97.1% | 62.9% | 14.3% | 0.0% |
| Time Critical Code | 35 | 97.1% | 44.1% | 14.7% | 0.0% |
| Time Critical Natural | 35 | 94.3% | 34.4% | 40.6% | 18.8% |

*Table 13.* Memoization semantic-vs-surface ablation on the canonical released grid: $N{=}35$ runs per phrasing aggregated across all 15 model configurations at $T{=}0$. Pass% uses the raw $N$ denominator; algorithm-family rates use the classified-output denominator (excluding unknown). All time/performance phrasings raise matrix-exponentiation use above the 0% baseline despite different surface forms, indicating that the semantic cue rather than typography drives the shift.

| Model | N | Pass% | Unknown% | Avg. capacity (pp) |
|---|---|---|---|---|
| DeepSeek-R1 70B (Llama-distill FP16) | 1,135 | 88.5% | 3.4% | 48.1 |
| DeepSeek-R1 70B (Llama-distill Q8_0) | 1,135 | 86.9% | 7.0% | 43.6 |
| Qwen2.5-Coder 32B Instruct (FP16) | 1,135 | 94.6% | 1.8% | 43.9 |
| Qwen2.5-Coder 32B Instruct (Q8_0) | 1,135 | 94.3% | 2.9% | 42.8 |
| DeepSeek-R1 70B | 1,135 | 89.3% | 1.4% | 45.5 |

*Table 14.* Quantization and reasoning-model summary at $T=0$. N is the per-configuration run count in the canonical dataset. Avg. capacity is the mean steering capacity (max–min pp across cue values, averaged over (task, cue-type) cells with $\geq 2$ cue values, classified denominator, `input_format` excluded for consistency with Table 2). DeepSeek-R1 Llama-distill Q8_0 lowers average capacity and pass rate relative to FP16; Qwen2.5-Coder 32B is essentially unchanged under Q8_0.

| Task | Cue | Value | GPT-5 | GPT-4o | Gemini | GLM | Claude |
|---|---|---|---|---|---|---|---|
| Expr. Parse | Docstring | none | recursive descent | eval | shunting yard | unk | recursive descent |
| Expr. Parse | Interaction | competitive time | shunting yard | shunting yard | unk | eval | recursive descent |
| LRU | Style | legacy | custom node | OrderedDict | OrderedDict | dict+DLL | unk |
| Memo | Ablation | time critical code | naive rec | space-opt | bottom-up DP | space-opt | matrix exp |
| Memo | Interaction | interview time | space-opt | space-opt | top-down | naive rec | matrix exp |
| Rate Limit | Review | conciseness | window | bucket | unk | window | naive |
| Rate Limit | Comment | moderate | window | naive | window | bucket | unk |
| Rate Limit | Ctx | teaching | window | unk | unk | bucket | naive |
| Rate Limit | Dep | stdlib only | window | naive | window | bucket | unk |
| Rate Limit | Dep | web | window | naive | window | bucket | unk |

*Table 15.* Top divergence cells across the five API models at $T=0$: each row is a (task, cue type, cue value) where the dominant algorithm family differs across four distinct labels (the maximum observed; no cell hit five distinct labels). Each cell aggregates 25 runs (5 replications $\times$ 5 API models); the entry reports the per-model argmax over classified outputs. `unk` indicates the per-model argmax is the unknown class.

