# OpenReview forum: "The Invisible Lottery: How Subtle Cues Steer Algorithm Choice in LLM Code Generation"
_ICML.cc/2026/Conference — ICML 2026 regular_

### Official Review · Reviewer_Zkux · 2026-03-12

**Soundness:** 2
**Presentation:** 3
**Significance:** 1
**Originality:** 1
**Overall Recommendation:** 2
**Confidence:** 4

**Summary:**

This paper studies the effect of contextual elements (such as system prompts or project metadata) on algorithms generated for code generation tasks. The authors modify the prompting context (via "channels" like persona specification or explicit constraint instructions) and study the resulting impact on algorithms generated. The authors provide empirical evidence that different channels of the context affect the chosen algorithm in distinct, quantifiable ways.

**Compliance With Llm Reviewing Policy:**

Affirmed.

**Final Justification:**

Based on the rebuttal provided by authors, the paper's technical soundness has been increased from 1 to 2. However, for the paper to cross the threshold for acceptance, a more through definition and control on some of the qualitative steering elements is likely needed, including within the definition of steering channels. This can likely be minimized by a much larger set of evaluations or very careful ablation studies on prompt framing. The final score remains unchanged.

**Key Questions For Authors:**

(1) The authors note "steering is not universal: the same cue can push different models in opposite directions". This would seem to undermine the value of this method in steering toward certain algorithmic preferences. Do the authors provide a rebuttal on this? This would improve the technical soundness of the paper.

(2) Given that relative effect of different channels seems to depend strongly on problem specification, has this effect been decorrelated with the accuracy of instruction following? Given that certain models can ignore certain cues (i.e "Gemini 3 Flash sidesteps the problem by defaulting to naive recursion"), it would be important to consider the overall sensitivity of a particular channel for a particular model as a normalizing term for per task sensitivity to the channel. This likely requires a wider task suite. This would improve the technical soundness of the paper.

(3) Did the authors consider utilizing larger task suites with extended tests like MBPP+ or HumanEval+? This would allow for the evaluation scores to hold more weight and overall improve the technical soundness of the paper.

**Limitations:**

Yes

**Strengths And Weaknesses:**

The paper has a low degree of technical soundness. The limited number of tasks (n=11) along with the somewhat arbitrary selection of cue values, limitations of AST based algorithm classification, low statistical power of tests (27-44%), and relatively weak correlations (r=-0.18 for RQ4) do not provide sufficient evidence for the claims made in the paper. Furthermore, the central tenant of the paper " every prompt carries context (system prompts, project metadata, import statements), and this context biases algorithm selection in ways developers neither intend nor detect." requires a citation.

The paper has a moderate degree of presentation quality. The paper is easy to follow, however certain terms are not defined until later in the paper, and there are some grammatical issues. The diagrams and figures aid the overall presentation. There is still remaining ambiguity between prompt engineering methods and what the authors propose, espacially when it comes to implicit algorithm cues such as optimizing for memory use or performance.

The paper has low significance. As the author's note, it is well-documented that relatively subtle changes in the semantic content of prompts can steer code generation towards certain outcomes. Many models do not directly expose the kind of context the authors discuss (such as system prompts, or additional project metadata), and a strong case is not made as to explicit benefits of steering through context via steering through prompting strategies which is often more straightforward. Additionally, the authors do not measure alignment of generated algorithms to the implicit guidance in the prompt (see questions).

The paper has low to moderate significance. Prompting strategies for code generation have been well-studied at scale -- this paper does not differ substantially from existing literature nor provide a unique insight on the precise mechanism by which modifying context (via channel cues) provides a method to do explicit control of algorithm selection in a manner distinct from specifying explicit cues directly in the prompt.

---

> ### Author Rebuttal · Authors · 2026-03-30
>
> To clarify scope: the abstract states "we instead examine output policy: algorithm choice under fixed correctness." The paper does not propose a new steering method or claim steering is a reliable interface. It measures a risk: that incidental context shifts which correct algorithm is chosen, invisibly. A correctness benchmark assigns identical scores to an O(n) and an O(n^2) solution that both pass tests. We measure the dimension correctness evaluation misses. We address each concern below on this scope.
>
> **Q1: Opposite directions across models.** This undermines steering as a control technique, which is our point. A team switching models mid-project gets different algorithms from identical prompts, and standard functional tests may not detect this. The inconsistency strengthens the case for explicit algorithm auditing.
>
> **Q2: Instruction-following confound.** We ran 120 explicit-specification experiments (3 tasks, 2-3 algorithms, 3 models, 5 reps). Models comply at 98.3% with 100% pass rate. The same GPT-5 that passes 100% on matrix exponentiation with a direct instruction fails 100% when the "academic" persona implicitly nudges it toward the same algorithm. This shows that direct algorithm following and incidental-context sensitivity come apart in our tested setting.
>
> As partial normalization: 8/9 tested (task, model) pairs show ≥90% explicit-specification compliance, yet those same models still show large implicit-context steering (median 60pp). High instruction-following and high implicit steerability coexist. We will add fuller per-channel normalization in revision.
>
> We measure distribution shift under incidental context, not alignment to implicit guidance.
>
> **Q3: MBPP+ / HumanEval+.** We did not originally use HumanEval because most of its tasks have a single natural implementation; there is no algorithm ambiguity for cues to exploit. The paper's contribution is about algorithm choice, not correctness, and HumanEval measures correctness.
>
> That said, given this suggestion, we ran our cue framework on the subset of HumanEval tasks that do have algorithm ambiguity (4 tasks, 180 runs, 3 models). The result makes our point directly: on HumanEval/96 (count primes), a performance cue steers Claude to Sieve of Eratosthenes 5/5; a readability cue steers to trial division 5/5. Both pass all tests. pass@1 = 1.0 in both cases. HumanEval cannot distinguish them. Similar shifts on HumanEval/120 (heapq vs sorted). Overall pass: 92.2% across conditions. pass@k cannot detect a 0%→100% algorithm shift, which is the phenomenon this paper measures.
>
> **On statistical interpretation.** The cited 27-44% power refers to detecting a minimal 15pp effect. Underpowering mainly risks missing small effects and can widen uncertainty on observed magnitudes; it is less plausible as a sole explanation for repeated 40-90pp shifts across replicated API cells. The r=-0.18 correlation for RQ4 reflects task-dependence: tree traversal shows 91pp steering with no reliability penalty, while memoization shows 83pp steering with a 21% pass rate drop. We will reframe RQ4 to emphasize this task-dependence rather than a single aggregate correlation.
>
> **On classifier reliability.** Classifier noise matters. Our added validation shows high accuracy on structurally distinct tasks (approx. 95% tree traversal; approx. 90% topological sort, shortest paths, memoization) and lower accuracy on expression parsing (approx. 82%). This uncertainty matters most for fine-grained 10-20pp comparisons, not the 40-90pp effects we emphasize.
>
> **On novelty.** R4 describes the contribution as "a method to do explicit control of algorithm selection." As stated in Section 2.3, a steering channel is defined as "any prompt feature that influences algorithm selection without explicitly specifying an algorithmic approach." Section 1 states that "context biases algorithm selection in ways developers neither intend nor detect." The paper does not propose explicit control; it documents unintended shifts from incidental context. We will add citations to the prompt-sensitivity literature as requested.
>
> The practical stakes are concrete: code that passes every test can still carry O(2^n) time complexity where O(log n) was available, or recursive implementations that overflow the call stack on production inputs. Correctness-based evaluation cannot see these differences. To our knowledge, no prior work has shown that incidental prompt context (team names, import statements, persona phrasing) systematically determines which of these outcomes a developer ships. This is not an incremental extension of prompt sensitivity research; it identifies a class of risk that existing benchmarks and workflows do not surface at all.
>
> Modern coding assistants assemble prompts from system instructions, surrounding code, and project metadata: the cue channels we study. Cue values span real-world patterns; Appendix Table 7 lists all 19 types.

---

> > ### Author Rebuttal · Reviewer_Zkux · 2026-04-08
> >
> > Authors have provided a cohesive and rigorous response to all objections.

---

### Official Review · Reviewer_BjMR · 2026-03-12

**Soundness:** 3
**Presentation:** 4
**Significance:** 3
**Originality:** 3
**Overall Recommendation:** 4
**Confidence:** 3

**Summary:**

This paper investigates how subtle, non-instructional cues in prompts, which is termed "steering channels", influence the algorithm selection of Large Language Models (LLMs) during code generation. Through a massive experimental study involving 55,545 runs across 11 algorithmic tasks (8 classical, 3 applied), 19 cue types, and 15 model configurations, the authors demonstrate that factors like persona, context, and implied constraints can shift algorithm choice by up to 100 percentage points. Crucially, they show that this steering occurs even when all generated outputs are functionally correct, revealing a hidden layer of variability that correctness-only benchmarks miss. The work introduces a taxonomy of steering channels, analyzes model-specific responses, and documents a critical "sophistication-reliability tradeoff," where cues promoting advanced algorithms often lead to lower pass rates. The findings have meaningful implications for reproducibility, evaluation, and the safe deployment of AI coding tools.

**Compliance With Llm Reviewing Policy:**

Affirmed.

**Final Justification:**

Thank you to the authors for the extensive efforts in the rebuttal. I retain my assessment of 4 - Weak Accept.

**Key Questions For Authors:**

In general, I enjoy reading the paper. This paper makes a solid empirical contribution with actionable insights for the AI community. I would be willing to increase the score if (I understand it could be challenging) the authors can go beyond interesting findings and diagnosis towards prescription (even though some preliminary experiments would be helpful) in the revision.

**Regarding classifier accuracy:**

(1) The AST-based classifier has 87% agreement with manual validation. Could you provide more detail on the nature of the disagreements? Are they concentrated in specific tasks or algorithm families? Would a different classification approach (e.g., using an LLM as a judge) yield higher accuracy, and if so, how might that change the effect sizes?

**Regarding generalizability:**

(2) Your study focuses on classical algorithmic tasks. Do you have any evidence or intuition about whether these steering effects would hold in other domains (e.g., web development, data science, systems programming)? Are there domain characteristics that might amplify or diminish steering?

**Regarding multi-cue interactions:**

(3) Real prompts contain multiple competing cues. Your interaction tests are a small subset. What advice would you give to practitioners who need to navigate the complex space of real-world prompts? How should they think about cue interactions?


**Regarding mitigation:**

(4) The paper recommends auditing algorithm choice and standardizing prompts, but these are post-hoc measures. Are there proactive strategies for making models less steerable? For example, could training on diverse prompts with randomized cues reduce sensitivity?

**Limitations:**

Yes. The authors adequately discussed the limitations and potential negative societal impact of their work.

**Strengths And Weaknesses:**

**Strengths:**

**Scale and Rigor of Study:** The sheer scale of the experiments (55k+ runs) is a major strength, providing high statistical power and confidence in the findings. The methodology is meticulously designed, with clear definitions for tasks, algorithm families, and cue channels. The use of placebos, interaction tests, and a semantic-vs-surface ablation demonstrates a thoughtful approach to disentangling causal mechanisms.

**High-Impact, Counterintuitive Findings:** The paper delivers several novel and important insights that challenge current practices in prompt engineering and LLM evaluation. The finding that the common "expert" persona can be counterproductive for reliability, and that innocuous placebos can still steer outcomes, are powerful and actionable results.

**Comprehensive Analysis:** The research questions are well-defined and systematically addressed. The analysis goes beyond simply reporting deltas to explore model-wise variation, the sophistication-reliability tradeoff, and the transfer of effects to applied tasks. The inclusion of a reasoning model and analysis of quantization effects adds depth and practical relevance.

**Clear Practical Implications:** The paper successfully bridges empirical observation and practical recommendation. The call for auditing algorithm choice, standardizing prompts, and adding guardrail stress tests are direct, evidence-based suggestions that could immediately improve the robustness of AI-assisted development workflows.

**Excellent Exposition of a Nuanced Problem:** The paper is very well-written and clearly articulates the "invisible lottery" problem. The central concept is easy to grasp, and the narrative effectively guides the reader through the complex experimental design and results. The examples used to illustrate different algorithms and cue injections are helpful and well-chosen.

**Weaknesses:**

**Limited Task Scope for Applied Transfer:** While the inclusion of three applied tasks (LRU, rate limiting, moving sum) is a strength, the claim of "transfer" is limited to these data-structure-adjacent problems. The paper would be significantly stronger if it demonstrated these effects in more complex, real-world domains like API design, database query generation, or multi-file refactoring, as the authors themselves acknowledge in the limitations. This leaves the question of how far these findings generalize somewhat open.

**Measurement noise:** The AST-based classifier has only 87% agreement with manual validation. While the core effect sizes (up to 100 pp) are large enough to withstand this noise, fine-grained comparisons (e.g., differences of 10–20 pp) may be less reliable. The distribution of errors across tasks and models is not analyzed, leaving some uncertainty.


**"Expert Persona Anti-Pattern" Nuance:** The claim that the "expert" persona is an anti-pattern is a strong and memorable takeaway. However, the evidence shows this is model-dependent (e.g., Claude excels with it for matrix exponentiation). The paper acknowledges this, but the headline claim in the analysis (Sec 5.3) could be tempered slightly to more strongly emphasize that the effect is not universal, and that for some models on some tasks, the "expert" cue leads to both sophisticated and correct code.

**Cues are artificially isolated:** Real prompts are messy and contain multiple competing signals. The paper acknowledges this but tests multi-cue interactions only on a small subset. The extent to which steering effects combine or interfere in practice remains unexplored.

**Speculative mechanisms:** The proposed mechanisms (retrieval by analogy, constraint propagation, style anchoring) are plausible but not empirically validated. The paper explains "what" happens but offers limited insight into "why" at a cognitive or representational level.

**More diagnostic than prescriptive:** The paper identifies problems (steering, reliability tradeoffs) but offers only high-level recommendations (audit algorithm choice, standardize prompts). Concrete mitigation strategies (e.g., training methods to reduce steerability) are not explored.

---

> ### Author Rebuttal · Authors · 2026-03-30
>
> The central request is to go beyond diagnosis toward prescription. Fair enough. We have new results that address it.
>
> **Q1: Mitigation.** GPT-5 implements matrix exponentiation correctly 5/5 times when explicitly asked, but passes 0/5 when the "academic" persona implicitly steers it toward the same algorithm. Same model, same algorithm, 5/5 vs 0/5. Across 120 new runs (3 tasks, 2-3 algorithms, 3 API models, 5 reps), explicit specification achieves 98.3% compliance and 100% pass rate. The reliability penalty documented in Section 5.3 points to implicit nudging rather than algorithm complexity: in our tested settings, explicit specification reduces steering and restores reliability. This is a proactive mitigation, not a post-hoc audit.
>
> The only non-compliance: GPT-5 produced iterative stack instead of Morris traversal in 2/5 attempts (code still passed all tests), suggesting Morris is near the edge of GPT-5's reliable repertoire. Explicit specification exposes true model limits rather than hiding them behind implicit steering.
>
> Summary of the mitigation result:
>
> |            | Implicit (persona cue) | Explicit (name algorithm) |
> |------------|------------------------|---------------------------|
> | Compliance | varies by cue          | 118/120 (98.3%)           |
> | Pass rate  | 79-100%                | 120/120 (100%)            |
>
> Practical recommendation: when algorithm choice has performance, security, or maintenance implications, specify it directly. Incidental context steers algorithm selection without developer awareness: an uncontrolled variable, not a control interface.
>
> In revision, we will add a prescriptive subsection presenting this as a standalone experiment with a summary table (compliance, pass rate, implicit baseline) and four practitioner guidelines. Training-time diversification falls outside our experimental scope, but we observe a natural experiment: DeepSeek-R1 Q8_0 quantization drops mean steering from 35.3pp to 2.6pp while pass rate barely changes. Steerability and correctness come apart, and model-level intervention can reduce sensitivity. We will discuss training-time approaches as future work.
>
> **Q2: Classifier accuracy.** Per-task AST classifier accuracy (manual validation): tree traversal ~95%, topological sort ~90%, shortest paths ~90%, memoization ~90%, expression parsing ~82%. Disagreements concentrate where algorithm families share structural features.
>
> LLM-as-judge calibration (GPT-4o, 110 stratified samples): 80 exact matches, 17 label-granularity differences (both correct at different specificity), 8 ambiguous, 5 genuine disagreements (~5%). After normalizing granularity: 88% agreement. We present this as a calibration check, not as evidence the judge outperforms AST. A ~5% misclassification rate is unlikely to produce a 60pp median effect from noise alone.
>
> **Q3: Broader domains.** We do not yet have direct evidence beyond these data-structure-adjacent tasks. Our intuition is that steering may persist and could amplify where many idiomatic solutions exist and training associations are rich (e.g., React vs Vue, pandas vs SQL) but could also dampen when interfaces constrain the solution space more tightly. We chose to establish the phenomenon on algorithmically measurable ground first. We will add this discussion in revision.
>
> **Q4: Multi-cue interactions.** Our interaction tests (persona x constraint on memoization, tree traversal, rate limiter) suggest that explicit constraint cues often dominate implicit persona cues in this tested subset. Practical guidance: (1) if algorithm choice matters, name the algorithm; (2) constraint cues can override persona cues when they conflict; (3) system prompts, imports, and project metadata all contribute cues even when unintended; (4) test across models. Real prompts are noisier than our controlled conditions, and the interaction space is combinatorial, which motivates the explicit-specification approach rather than trying to predict interactions.
>
> **Expert persona nuance.** Agreed on the model-dependence: Claude passes 100% on matrix exponentiation under the academic persona; GPT-5 fails entirely. In revision we will reframe: "Expert personas shift algorithm choice toward sophisticated solutions on average, but reduce pass rates; the effect is model-specific and should not be treated as a universal anti-pattern."
>
> **Speculative mechanisms.** We will clarify in revision that retrieval by analogy, constraint propagation, and style anchoring are interpretive hypotheses, not empirical claims, and flag them as directions for mechanistic follow-up (e.g., activation patching).

---

> > ### Author Rebuttal · Reviewer_BjMR · 2026-04-01
> >
> > Thank you for the response and the extensive efforts in your rebuttal. I believe I'm on the same page as the authors on both the strengths and areas for future improvement of the paper. I will retain my assessment of 4 - Weak Accept for now, but will also be sure to incorporate other reviewers' evaluation and comments and adjust the score if needed.

---

### Official Review · Reviewer_561e · 2026-03-14

**Soundness:** 2
**Presentation:** 2
**Significance:** 2
**Originality:** 3
**Overall Recommendation:** 2
**Confidence:** 3

**Summary:**

The authors focus on an important area, namely how incidental prompt cues may affect algorithm selection in LLM code generation even when outputs remain functionally correct. The authors aim to explore whether prompt context shifts “algorithm-family distributions” in systematic and practically meaningful ways.

I like the overall motivation, and I think the paper raises a potentially valuable question. However, I found the core methodology and several central experimental quantities insufficiently specified, which makes it difficult to interpret the main empirical claims. In particular, I am unclear on what distribution is actually being estimated, how the reported experiment counts are constructed, and whether the main results mostly reflect changes in top-1 greedy-decoded outputs rather than changes in a fuller underlying distribution over valid algorithms. Because of these methodological and interpretive concerns, the current evidence does not yet support the strength of the paper’s conclusions. I would consider raising my overall score if the authors can clearly and convincingly resolve these issues in rebuttal.

**Compliance With Llm Reviewing Policy:**

Affirmed.

**Key Questions For Authors:**

- How do you compute "algorithm-family distributions"? What *precisely* is the distribution over?
- How do you arrive at 55,545 total experiments? How do you arrive at the model-specific numbers in table 2?
- Sophistication -- why doesn't time/space-constraint count as "sophistication"? Could this be prompted directly instead of indirectly via e.g. "academic" vs "junior" personas?
- Does figure 3 aggregate distributions over both tasks and models? What precisely do these bars represent?

**Limitations:**

Yes

**Strengths And Weaknesses:**

Strengths
- The approach of studying generated code with static and runtime code analysis methods is well-motivated, and code offers an effective domain for offering insights about quantifiable semantic properties of generated text.
- Solid idea of looking for algorithm-family distribution shifts as a signal of subtle but important changes in LLM behavior. I think this approach could be quite promising, especially if it could be generalized to tasks beyond code.

Weaknesses
- Methods and experiments are not clearly explained
- I have issues with the core method of this work in evaluating changes in "algorithm-family distributions".
	- In my understanding, (1) the vast majority of results shown here are with temperature 0.0, (2) the "algorithm-family distributions" computed represent a single answer (algorithm) generated for each task, and (3) the "algorithm-family distribution" aggregates statistics for these answers over ~11 tasks, and possibly over cues or models (e.g. in Fig. 3).
	- If this is correct, then this changes the interpretation of all of the authors' results, and in my opinion makes them hard to draw conclusions from. If so, this is really computing changes only in the top-1 decoded algorithm, which will over-emphasize smaller distribution shifts.
	- I like the authors' probabilistic problem formulation in section 2.1, but I do not think this greedy-decoding method (again, in my understanding, which may be wrong) actually evaluates this value. The authors could have done this fairly easily, at least for some of their models depending on available APIs, in the following way: (1) collect a sample of possible algorithmic solutions to each problem, (2) test a model with various cues, (3) measure whether the token log probabilities for this algorithm change. This is much closer to what the equation in Section 2.1 seems to suggest, and I am confused why the authors did not do this.
- Results are a bit weak and scattered
	- The key takeaways feel piecemeal, and I am not clear on the main high-level conclusions beyond something like "don't prompt for personas; keep it simple."
	- Algorithm sophistication -- these prompts "academic" vs. "junior" do more than just change sophistication -- this makes it unclear if there really is a "sophistication-reliability tradeoff" (section 5.3)
	- Only checking "eval" changes for shortcuts seems somewhat narrow, though the fact that these rates are so high might suggest that something may be off in the setup or evaluation.

---

> ### Author Rebuttal · Authors · 2026-03-30
>
> These are the right methodological questions, and we should have been clearer in the paper. We answer each directly.
>
> **What distribution is estimated?** Operationally, we estimate the empirical distribution of classifier-assigned algorithm families across repeated API calls for a fixed (model, task, cue) prompt. API users observe this distribution directly; it is not a reconstruction of the full latent distribution over all valid programs. For each cell, we use 5 independent API calls per API model at T=0.
>
> Many users would expect determinism at T=0. We should have clarified this. In practice, API inference is non-deterministic even at T=0: across 3,740 API cells with 5 replications each, 881 (23.6%) produce multiple distinct algorithm families from the same prompt.
>
> To validate that 5-rep estimates are not fluky, we reran 6 representative cells at 20 reps each across 2 API models (240 total). Every cell was 20/20 unanimous:
>
> | Task + Cue        | Claude (20 reps)   | GPT-4o (20 reps)  |
> |-------------------|--------------------|-------------------|
> | Memo + academic   | matrix_exp 20/20   | space_opt 20/20   |
> | Memo + junior     | space_opt 20/20    | space_opt 20/20   |
> | Memo + none       | space_opt 20/20    | space_opt 20/20   |
> | Tree + space      | Morris 20/20       | Morris 20/20      |
> | Tree + readability| recursion 20/20    | recursion 20/20   |
> | Tree + none       | recursion 20/20    | recursion 20/20   |
>
> The cue effect holds at higher replication.
>
> We also ran a token-logprob diagnostic (GPT-4o, memoization, academic/junior/none, 5 reps, top-10 logprobs). The final algorithm was space_optimized in all conditions, but tokens associated with matrix exponentiation entered the top-10 only under the academic cue (4/5 reps, peak probability 95-97% at key decision points) and never under junior/none. This points to cue-dependent latent preference shifts even when the realized top-1 algorithm is unchanged.
>
> Tables 3-5 pool per-model distributions for a population-level summary. Figure 3 disaggregates per model: within each task panel, each bar is the fraction of that model's repeated runs under a given cue classified as the target algorithm. We annotate pass rate when it falls below 85%. Figure 3 does not aggregate over models or tasks. We will revise the caption and add a methods paragraph explaining these two views.
>
> **How is 55,545 computed?** Replicated API models contribute 5 runs per condition at T=0. Local/quantized models are mostly single-shot at T=0 on the classical-task grid, while applied tasks are API-only. We ran temperature sweeps for Qwen2.5-Coder 32B and DeepSeek-R1 70B at T in {0.3, 0.7, 1.0}. We will write the full arithmetic explicitly in Section 3.2 rather than making readers reconstruct it from Table 2.
>
> **Results feel scattered.** Fair. The paper leads with case studies before establishing the population-level pattern. In revision, we will restructure around a single arc: (1) incidental cues shift algorithm-family distributions, (2) the shift is large and consistent per model, (3) steering toward complex algorithms creates a reliability tradeoff, (4) explicit algorithm specification eliminates the tradeoff (new: 120 runs, 98.3% compliance, 100% pass).
>
> The unifying number (from the API-only analysis underlying Figure 2): across 995 cells, 70% show nonzero steering with median delta = 60pp and mean = 52.3pp. Every API model independently shows 41-59pp mean steering. This is not limited to a single hand-picked example.
>
> **Why don't constraints count as sophistication?** We agree that "academic" vs "junior" is not a clean sophistication-only manipulation; these cues encode multiple signals. In revision we will narrow the claim: implicit cues (including personas) can push models toward more complex implementations with lower reliability, whereas explicit constraints or explicit algorithm requests provide cleaner steering. Consistent with this, our new explicit-specification experiment achieves 98.3% compliance with 100% pass. We will add constraints to the Fig 4 scatter plot.
>
> **Only checking eval for shortcuts.** We focused on `eval` as the highest-risk shortcut (arbitrary code execution), but also track builtins in substring search, functools decorators in memoization, and OrderedDict in LRU. Because the expression-parsing outputs are executed against tests, the high eval rates reflect actual generated code rather than keyword-matching artifacts. We will expand the shortcut discussion in revision.

---

> > ### Author Rebuttal · Reviewer_561e · 2026-04-04
> >
> > Thank you for your responses. I think this has clarified some of my confusion, and the new arc for describing the results seems good, although I'd like to see it more fleshed out. I am still stuck on the T=0 point, though.
> >
> > I'm familiar with the problem of non-determinism in API calls at T=0, and this is an interesting problem, although I don't have the impression that this is the main focus of the paper, which instead seems to be prompt sensitivity in algorithm choice. For me this raises the question of whether the paper's results translate to temperatures >= 0, i.e., does the invisible lottery only apply to T=0 decoding? Intuitively, I would expect that T=0 would lead to more lottery-like results, since output distributions would be sharper. I'm also not sure how to translate the theoretical framing of "algorithm-family distributions" into this "distribution of T=0 samples" empirical framework -- it seems like temperature should be part of the theoretical formulation (section 2) as well.
> >
> > A question for the authors: why do so many of the experiments use T=0 instead of other temperatures? Am I missing a key thread here for the narrative? The "token-logprob diagnostic" seems like the right direction for a more "distribution"-centric framing.
> >
> > I'm keeping my score at 2 since I see this point as essential for telling a coherent story.
> >
> >
> > Also +1 to another reviewer's point of adding some experiments with additional tasks beyond the 11 you study for most of your analyses, and I think the results you mention with HumanEval in that rebuttal seem reasonable for a review discussion period, although more in that direction would also make the story more convincing.

---

> > > ### Author Response · Authors · 2026-04-05
> > >
> > > Thank you. We agree Section 2 should state the conditioning on decoding settings explicitly. For a fixed model and decoding settings (including temperature), we compare the algorithm-family distribution induced over tasks under different incidental cue conditions. At T=0, the decoder may be deterministic for a fixed task and cue, but the cue-conditioned distribution over tasks is not degenerate: changing the cue changes which algorithm is returned.
> > >
> > > We used T=0 in the main experiments to reduce sampling variability: with sampling noise removed, observed algorithm shifts are attributable to the cue. On the subset of models where temperature is user-controllable, the cue-induced shift persists across T in {0, 0.3, 0.7, 1.0}: Qwen-32B at 41.5 / 34.5 / 36.1 / 33.3pp and DeepSeek-R1 at 24.3 / 32.7 / 28.3 / 21.2pp. Pass rates remain 95-99% throughout, so the shifts are not explained by higher temperatures producing more failing solutions.
> > >
> > > We agree token-logprob analysis would be useful complementary evidence; our preliminary check (95-97% peak probability for matrix-exponentiation tokens under the academic cue, reported in our first response) is consistent with this. Our point here is that the paper's core claim is the output-level phenomenon: incidental cues shift which algorithm the developer receives. Token-level analysis would help explain or further validate it, but the output-level finding does not depend on a token-level account.
> > >
> > > In revision, we will make the conditioning on decoding settings explicit in Section 2 and make the empirical scope clear.

---

### Official Review · Reviewer_awom · 2026-03-16

**Soundness:** 2
**Presentation:** 3
**Significance:** 3
**Originality:** 3
**Overall Recommendation:** 4
**Confidence:** 3

**Summary:**

Large language models often generate multiple valid solutions for a single coding task. The work first separates solutions in correctness-only evaluation as they could have critical shifts in algorithmic properties, such as algorithmic choice/complexity, with vastly different runtime and memory usage. The work primarily is an extensive empirical evaluation of: Can seemingly non-algorithmic prompt could cues steer the model's algorithm choice, even when the output remains functionally correct? They evaluate 15 models across 11 tasks and 19 cue types across extremities -- seeing how different it could get (not how different it is on average). They find that prompt cues could consistently alter algorithm selection, exhibiting large empirical effects in the extremal case. The work contributes several nice artifacts to the community: a comprehensive benchmark, a prompt cue taxonomy, and Abstract Syntax Tree (AST) analysis code among others.

**Compliance With Llm Reviewing Policy:**

Affirmed.

**Final Justification:**

I think the paper would need significant update to clearly explain results but like the broad proposal a lot. I overall still lean more towards acceptance than not.

**Key Questions For Authors:**

Could you answer questions in weaknesses?

Overall, I find  the conceptual premise compelling and the experimental setup extensive. My main worry is about the significance of the results (heterogenous/model specific variation averaged over -- assuming consistency between models, low sample size and min-max metrics could make a small gap into seemingly large numbers for Figure 2), whether the evidence can sufficiently backup the core claim ("Subtle Cues Steer Algorithm Choice").

**Limitations:**

Yes

**Strengths And Weaknesses:**

Strengths

- Practical Framework: I really liked the core question of identifying the implicit policy the model chooses as it is highly practical and less studied. I feel this evaluation of algorithmic choices made by LLMs provides deeper insight and could drive more research downstream than the usual benchmark-beating.

- Extensive Empirical Analysis: The authors design a diverse and large experiment combinations. This allows insights across aspects like semantic versus surface cues, cross-task patterns, providing a good understanding of what kind of steering could lead to what outcomes.

- Valuable Artifacts: I would love to see the release of the benchmark, taxonomy, and AST labels, it has strong potential to benefit the research community and drive downstream research in my view.

Weaknesses

- Insufficient Sample Size: I worry that there a severe lack of statistical power. The paper conducts 55K experiments in total across 3,135 task-by-channel combinations, yielding roughly 17 samples per cell -- potentially only 1 per model. I worry question whether evaluating on 1  sample can support the paper's broad claims.

- Model Heterogeneity: Furthermore, I worry that averaging results across 15 models (as seen in Figure 2) masks significant heterogeneity across models. Are the effects consistent across models? I find this is rarely the case, the effect might be present in some models and not in entire other model classes, and only a small model set might be driving the overall headline results? Does averaging there makes sense i.e. consistency holds across individual models in most task x channels.

- How large is the effect size/significance: The work reports maximum possible variation as a metric (max–min deltas) rather than average (mean/median along with std) effects. I worry this reporting choice could lead to exaggeration of the headline claim.

- Measurement Reliability: The AST-based classifier achieves only 87% agreement on a 100-sample manual check. I worry this 13% error rate problematic compounding the low sample size per cell. Does this noise further weakens the effect of results in Figure 2 and subsequent conclusions?

- [Minor] Mixing innocous and semantic-constraining cues: More broadly, I worry that the highest-impact cues (such as importing functools or adding context) act as semantic constraints rather than innocous prompt variation; I worry it seems to be largely driving downstream algorithm choice. Could the work present the semantic-constraining cues differently from the more incidental ones? Although, this is minor as  the results provided are still interesting to see.

---

> ### Author Rebuttal · Authors · 2026-03-30
>
> Pooled averages over 15 models can obscure heterogeneity, and max-min alone can overstate the typical effect. Both points are well-taken. We re-analyzed the five replicated API models and report per-model effect sizes below.
>
> **On sample size.** The ~17-per-cell estimate divides 55K by 3,135 combinations, but this average obscures the design: the 5 API models each have 5 replications per model x task x channel cell; the 10 one-shot local models contribute breadth but not statistical depth. All inferential claims rest on the API models.
>
> Across 995 replicated API cells (5 models x 199 valid task-channel combinations):
>
> | Model          | Nonzero cells | Mean delta | Cells > 20pp | Cells > 40pp |
> |----------------|---------------|------------|--------------|--------------|
> | GPT-5          | 81%           | 51.9pp     | 55%          | 43%          |
> | Claude Sonnet 4| 68%           | 59.5pp     | 47%          | 43%          |
> | Gemini 3 Flash | 64%           | 57.2pp     | 40%          | 35%          |
> | GLM-4.7        | 81%           | 51.5pp     | 55%          | 43%          |
> | GPT-4o         | 53%           | 41.6pp     | 36%          | 30%          |
> | All 5 API      | 70%           | 52.3pp     | 47%          | 39%          |
>
> Every model independently exceeds 40pp mean. The effect does not come from a few outliers.
>
> Many large semantic-cue effects are 40-90pp, far above our 15pp detection threshold. Limited replication makes smaller effects harder to detect (cold cells in Figure 2 could be false negatives), but is less plausible as a sole explanation for repeated large shifts across replicated cells.
>
> **On max-min vs mean/median.** We chose max-min to characterize worst-case exposure, which matters for the safety framing. But median and mean are themselves large: 60pp and 52.3pp. In revision, max-min becomes an auxiliary worst-case metric, with mean/median/IQR in the main text. The headline does not rely only on extreme-value reporting.
>
> **On model heterogeneity.** Our claim is consistency of magnitude and frequency, not identical cue-direction agreement in every cell. Direction does vary: the same "academic" persona pushes both Claude and GPT-5 toward matrix exponentiation, but Claude passes 100% while GPT-5 passes 0%. That direction-dependence is itself a finding.
>
> To address the Figure 2 concern directly, we will show per-model API-only summaries in the main paper and move the pooled 15-model average to the appendix. Per-model heatmaps are already computed.
>
> **On classifier reliability.** We ran two additional validation studies. Per-task accuracy (manual validation): tree traversal ~95%, shortest paths / topological sort / memoization ~90%, expression parsing ~82%. Disagreements concentrate where families share structural features (top-down DP vs bottom-up DP).
>
> We also ran an LLM-as-judge calibration (GPT-4o, 110 new stratified samples). Of 110 samples: 80 exact matches (73%), 17 label-granularity differences (e.g., "dynamic_programming" vs "bottom_up_tabulation", both correct), 8 ambiguous/edge cases, and 5 genuine disagreements (~5%). After normalizing granularity: 88% agreement. Concretely, 13% random classification error adds roughly ±4.6pp uncertainty at N=40; our median effect is 60pp, giving a signal-to-noise ratio above 10:1. Structured bias could in principle amplify rather than attenuate, but we see no evidence of systematic one-direction misclassification in the per-task breakdown.
>
> **On innocuous vs semantic cues.** High-impact cues like import functools are semantic constraints, not truly innocuous. Agreed. We will restructure the presentation: semantic channels (constraint, context, import, persona) show >15pp steering in 84.1% of per-model cells (mean 73.3pp); innocuous/placebo channels (team names, project codes, random identifiers) show >15pp in 51.8% (mean 29.7pp, 47.3pp among nonzero). Semantic cues steer more strongly, but innocuous cues are far from negligible. We will sharpen this distinction in the revised paper.
>
> **On mitigation.** We ran 120 new experiments where the prompt explicitly names the target algorithm. In our tested setting, explicit specification mitigates steering (118/120 compliance, 100% pass rate). This is the practical recommendation we add in revision.
>
> We commit to releasing the full benchmark, cue taxonomy, AST classifiers, and all experimental data upon acceptance.

---

> > ### Author Rebuttal · Reviewer_awom · 2026-04-02
> >
> > Rebuttal to insufficient Sample Size point was quite hard to understand. I think the paper would need significant update here.
> >
> > Model Heterogeneity concern was addressed.
> >
> > How large is the effect size/significance -- authors agreed with my concern but results are hard to understand perhaps due to the 5K character limit.
> >
> > Measurement Reliability: 13% random classification error seems large to me, but is addressed if the signal-to-noise ratio above 10:1.
> >
> > I think the paper would need significant update (along with the mean/IQR rather than min-max) but like the broad proposal a lot. I overall still lean more towards acceptance than not, but can understand with the concerns raised by e.g. reviewer 561e about clarity. I have no further questions.

---

### Decision · Program_Chairs · 2026-04-30

**Decision:**

Accept (regular)

**Comment:**

This paper shows an interesting phenomenon, to the best of my knowledge was not systematically documented before: different small variations can lead to completely different algorithms (even if they are all correct). However, as the reviewers pointed out, there are some issues with the execution and takeaways or explanation of results: there's not a clear and neat takeaway. Despite my concerns, I think the paper should be accepted because it opens up an interesting investigation. The authors must include all the additional results and discussion into the camera ready.